# Cultural variation in young children's social motivation for peer collaboration and its relation to the ontogeny of Theory of Mind

Roman Stengelin[1,2]*, Robert Hepach[1,3,4], Daniel B. M. Haun[1,2]

**1** Department of Comparative Cultural Psychology, Max-Planck-Institute for Evolutionary Anthropology, Leipzig, Germany, **2** Leipzig Research Center for Early Child Development, Leipzig University, Leipzig, Germany, **3** Department of Research Methods in Early Child Development, Leipzig University, Leipzig, Germany, **4** Department of Experimental Psychology, University of Oxford, Oxford, United Kingdom

* roman_stengelin@eva.mpg.de

**Data Availability Statement:** All relevant data are within the manuscript and its Supporting Information files.

## Abstract

Children seek and like to engage in collaborative activities with their peers. This social motivation is hypothesized to facilitate their emerging social-cognitive skills and vice versa. Current evidence on the ontogeny of social motivation and its' links to social cognition, however, is subject to a sampling bias toward participants from urban Western populations. Here, we show both cross-cultural variation and homogeneity in three- to eight-year-old children's expressed positive emotions during and explicit preferences for peer collaboration across three diverse populations (urban German, rural Hai||om/Namibia, rural Ovambo/Namibia; $n$ = 240). Children expressed more positive emotions during collaboration as compared to individual activity, but the extent varied across populations. Children's preferences for collaboration differed markedly between populations and across ages: While German children across all ages sought collaboration, Hai||om children preferred to act individually throughout childhood. Ovambo children preferred individual play increasingly with age. Across populations, positive emotions expressed selectively during collaboration, predicted children's social-cognitive skills. These findings provide evidence that culture shapes young children's social motivation for dyadic peer collaboration. At the same time, the positive relation of social motivation and social cognition in early ontogeny appears cross-culturally constant.

## Introduction

Ranging from small hunter-gatherer groups to industrialized urban societies—humans depend on collaborative social interactions in which two or more individuals pursue joint goals interdependently [1]. Interdependence between collaborators demands mental perspective-taking and coordination because individuals have to monitor and adapt their actions, perspectives, beliefs, and goals to those of their counterparts [2, 3]. Engaging in collaboration thus offers a unique learning context for the consolidation of social cognition.

Young children across diverse populations are capable and competent in collaborating with their peers for mutual benefit [4–12]. In urban, Western societies, children start to collaborate

**Funding:** The work was funded by internal budgets of the Department of Early Child Development and Culture at Leipzig University. No other parties than the authors were involved in study design, data collection, and preparation of the manuscript.

**Competing interests:** The authors have declared that no competing interests exist.

under adult supervision within their second year of life and transfer this skill to successfully master peer interactions in the years to follow [4, 13]. In such contexts, peer collaboration often takes place in dyadic, playful settings in which children are encouraged to autonomously choose their social partners based on their preferences [14, 15]. In societies in which social relatedness is prioritized over autonomy, children's proclivity for collaboration is no less ubiquitous. Some studies suggest that children from rural non-Western, traditional populations may be even more skilled in coordinating and solving collaborative tasks than their urban Western counterparts [6, 10, 16]. For example, Rogoff and colleagues showed that children of Mexican descent growing up in the U.S. outperform their counterparts of European descent in some coordinative aspects of peer collaboration [10, 16]. That is, children with Mexican indigenous backgrounds coordinate their behaviors with their peers' by building upon the partner's actions fluidly and non-verbally. Children of European descent do so more often by relying on verbal coordination and parallel engagement. While these studies thus indicate cultural differences in how children collaborate with their peers, it is without much doubt that the mastery of peer collaboration constitutes a central task in young children's social development [4, 17, 18].

It has been posited that the early emergence of collaboration in human ontogeny relies on children's social motivation to interact [2, 17, 19]. Evidence in support of this notion mostly stems from urban Western populations. For example, 4- to 11-year-old U.S.-children express more positive emotions (e.g., intensity and frequency of smiles) when solving problems in collaboration with a peer as compared to solving them individually [20]. Furthermore, 3-year-olds from urban Germany prefer to work collaboratively rather than solitarily to obtain rewards [11]. However, the degree to which the social motivation for seeking and liking peer collaboration ([11, 20], see also [21]) is present outside urban Western populations is currently unclear [22, 23].

In urban Western populations, children's social development within social interactions is typically scaffolded and supported by parents and other caregivers. Adults support children's learning from and within social interactions by praising them for socially competent behaviors [14]. Children are free to engage with others based on their preferences [14] and learn to collaborate with others under adult supervision [13]. By age 2, children begin to coordinate activities themselves and increasingly engage in peer collaboration without external supervision [13, 24]. From this age onwards, dyadic peer collaboration is ubiquitous in children's everyday social interactions among urban Western populations. However, this practice of actively shaping and rewarding children's social interactions with peers is not representative of children's socialization across societies.

In many traditional farming communities, for example, parents expect their children to learn from them by observing them and through active participation in daily activities [25, 26]. Here, children mostly engage with peers, rather than adults, playfully, even though play is typically embedded into daily chores and practices [27]. Compared to urban, Western populations, parents incentivize and reward children's socially appropriate behaviors and contributions less actively. Instead, children rely more on their interest to navigate interactions with peers and adults to learn and participate. Here, caregivers often value hierarchical relatedness, obedience, and conformity as central socialization goals [28, 29]. The Ovambo, for example, a Namibian agro-pastoralist population, emphasize these values in their childrearing practices [30–32]: Children's relatedness to the social group is emphasized by caregivers' use of directive and assertive communication strategies [30, 31]. Young children are frequently tasked with household duties and demanded to exercise these autonomously and without much adult supervision.

In traditional hunter-gatherer populations, such as the Hai||om of northern Namibia, caregivers value autonomy as an essential socialization goal [33, 34]. Children are typically free to

structure their activities without substantial social obligations toward others, allowing more opportunities for individual rather than collaborative learning. Adults and children in hunter-gatherer communities may even actively avoid subordination and dependence between individuals [35]. Compared to urban, Western societies, the Hai‖om give little emphasis on child-centered pedagogy, and adult-supervised collaboration is rare among Hai‖om parents because of their cultural emphasis on individual autonomy. In contrast to Western populations, adults rarely praise children for socially competent behaviors but rather rebuke them when deviating from social norms [35, 36].

In sum, children from diverse populations engage in peer collaboration, even though populations show striking and systematic differences in how social interactions are framed culturally. In Western industrialized societies, such as urban Germany, adult caregivers typically consider play-like activities as an essential experience for their children. Notably, such experience is often detached from social obligations and family chores [37]. Here, children flexibly choose with whom they want to interact, which reflects the cultural emphasis on psychological autonomy over action autonomy [29, 38]. In consequence, social interactions can be entered and dissolved depending on whether they are experienced as personally rewarding or not. In traditional farming societies, such as the Ovambo of Namibia, a mixture of social relations and action autonomy frames children's socialization experience. Among traditional foraging societies, such as the Hai‖om of Namibia, individual autonomy is emphasized in both the psychological and the action domain [29, 33]. It is, however, unclear whether and how such variation in socialization goals affects children's social motivation to engage in peer collaboration, given that the vast majority of studies in this area have been conducted in urban Western populations [11, 17, 20]. While social motivation has been hypothesized to be an ontogenetic driver of social-cognitive development in Western societies [2, 3, 17, 39], it remains unclear whether this link can be generalized across populations.

To address this issue, we assessed both phenomena in 3- to 8-year-old children from 3 different populations. Children from urban Germany ($n = 80$) were assessed as proxies for Western urban populations. Hai‖om children ($n = 80$) from two rural populations in Namibia and Ovambo children from rural Namibia ($n = 80$) were examined to investigate the differential effects of cultural emphasis on children's autonomy, interpersonal relatedness and action autonomy on children's social motivation [29, 40]. Dyads of children (matched for sex and age) played a game in which they could obtain rewards from a device by pulling ropes. In a within-subject design, children retrieved the rewards either through individual action or through peer collaboration (8 trials in total). As the first measure of children's social motivation, we coded their positive emotional expressions during each trial. In addition to the gaming context (individual or collaborative), we manipulated the value of the rewards (high reward or low reward). We included this factor to validate whether our coding scheme would reliably identify changes in children's emotional expressions as a function reward value. As a second measure of social motivation, we investigated children's explicit preferences by letting them choose to obtain a reward either individually or collaboratively.

We hypothesized that if children's social motivation to collaborate was a cultural regularity, participants across all three populations should show a bias toward collaboration, as indicated by both, more positive emotional expressions and a preference for collaboration. If, in contrast, this motivation was culturally variable, the three populations should show different patterns: In line with previous studies, German children should show a strong social motivation due to frequent early adult scaffolding of collaborative behaviors and the praise received in social interactions [11, 20]. Hai‖om children should show a lower motivation to collaborate due to the cultural-typical emphasis on child autonomy [35]. The limited role of adult supervision encouraging and praising young children during social interactions among the Hai‖om

could also lower children's social motivation for peer collaboration. Our expectations regarding Ovambo children were mixed. On the one hand, the limited supervision and praise among Ovambo caregivers might reveal a similar pattern as predicted for Hai‖om children. On the other hand, the local emphasis on interpersonal responsibilities and social relations might also foster children's motivation to collaborate. In the case of revealing cross-cultural variability in children's social motivation, we further predicted that such differences should become more pronounced with increasing age.

According to the *Social Motivation Theory of Autism* [2], but also frameworks posited by other scholars [17, 41], children's tendency to seek and like social interactions (i.e., their social motivation) functions as an ontogenetic foundation for social-cognitive development: Children who are motivated to engage with others more frequently and persistently and thus spend more time in interactions in which they learn to understand and predict others' behaviors based on social-cognitive inferences. Collaborative endeavors, in which different partners pursue a joint goal, comprise a particularly important context in this regard given that social interdependence and the necessity to coordinate urge individuals to track the beliefs, thoughts, and actions of their collaborative partners [3]. Studies have provided initial evidence that social motivation may indeed be linked to Theory of Mind development among urban Western children (e.g., [42]). Yet, it remains unclear whether this link can be applied outside such populations. First, recent research has revealed that children's Theory of Mind skills vary considerably across cultures (e.g., [43–47]). Moreover, other studies showed that psychological correlates of Theory of Mind typically observed among Western populations do not necessarily persevere outside such samples. For example, while the number of siblings is a well-documented predictor of Theory of Mind among Western children [48], this effect is not evident among Iranian children [46, 49, 50]. Similarly, authoritarian parenting practices are negatively linked to U.S.-American children's Theory of Mind acquisition, but such links are absent among Korean children tested in the same study ([50], see also [51]).

To test the cross-cultural regularity of the hypothesized links between social motivation and Theory of Mind, we used an adaption of a Theory of Mind scale that has previously been applied cross-culturally [52]. In a series of tasks, children were asked questions about the mental states of fictive characters. We used a composite score of children's performances across tasks as a proxy for their Theory of Mind skills.

We expected inter-individual variation in both positive emotional expressions and explicit choices to be linked to children's Theory of Mind regardless of population and age since the interplay between social motivation and social cognition has been assumed to be recurrent across populations [2, 3]. If, in contrast, such links would vary across populations, this would call assumptions on the regularity of social motivation as a driver of social cognition into question.

## Methods

### Participants

A total of 240 children (120 dyads, $M_{Age}$ = 5.87 years, $SD_{Age}$ = 1.18, range = 3.54 to 8.35) participated in the study. In both Namibian populations, children were recruited via an opportunity sampling approach to maximize the number of participants in the respective communities. German children were recruited from an online database after obtaining written consent from their parents. We chose to assess children within this age range to ensure that children would be capable of collaborating flexibly with their peers while being sufficiently motivated and challenged by the endeavor. We paired each participant with a familiar same-sex peer of approximately similar age ($M_{Age\ Difference}$ = 0.45 years, $SD_{Age\ Difference}$ = 0.58). Children within

each dyad were familiar with each other. Participants came from three different populations: Forty German dyads ($n$ = 80, 42 girls, $M_{Age}$ = 5.77, $SD_{Age}$ = 1.22, $M_{Age\ Difference}$ = 0.30) participated in a mid-sized German town. Children in this population typically attend institutional childcare and participate in institutionalized education from their second to third year of life onwards. Children were tested in local kindergartens or primary schools. 40 Hai‖om dyads ($n$ = 80, 40 girls, $M_{Age}$ = 6.11 years, $SD_{Age}$ = 1.02, $M_{Age\ Difference}$ = 0.68) from two rural villages in Northern Namibia also participated in the study. Children in these villages attend local primary schools from around six years of age onwards. Finally, 40 Ovambo dyads ($n$ = 80, 38 girls, $M_{Age}$ = 5.73 years, $SD_{Age}$ = 1.26, $M_{Age\ Difference}$ = 0.37) were tested in a small town in Northern Namibia. All Ovambo children attended either a local kindergarten (typically starting from around two to three years of age) or a local primary school. Five additional Ovambo dyads were tested but excluded from the analyses because children were later found to be older than our targeted maximum age of eight years (3 dyads) or because one or both children were later found to have grown up in a non-Ovambo household (2 dyads). Among Hai‖om participants, one additional dyad did not finish the study because one child did not want to participate further.

The study design of this project was approved by [blinded for review], the Ministry of Home Affairs and Migration of the Republic of Namibia, the Regional Council of Oshikoto Region in Namibia, and the Working Group of Indigenous Minorities in Southern Africa (WIMSA). Parental consent (verbal or written, depending on parents' literacy) and school principal's signed consent was obtained before children participated in the study. All dyads were tested in quiet rooms in local schools or daycare centers. Participation was strictly voluntary.

## Materials

**Collaboration task.**   Each child sat on a cushion (diameter = 34 cm) placed on the floor in approximately 2m distance from the devices. Instructions came from a 35-slide multimedia presentation (Microsoft PowerPoint) presented on a laptop screen (MacBook Air, 13"). Audio-files were translated from English into the respective language (Oshindonga/Oshikwanyama, Hai‖om, and German) by native speakers, and another native speaker checked the translation afterward. Disagreements between translations were minor and solved through discussion among the two translators and the first author. Audio files of the final translations were subsequently recorded and embedded in the videos of the presentation. The arrangement of devices presented on the laptop screen was identical to that in the testing room. We used two types of wooden devices (30 cm x 15 cm x 11 cm) to manipulate either collaborative or individual efforts for obtaining rewards (following [5]). Within each device, children could move a wooden block (5 cm x 10 cm x 5 cm; colored either blue or red) by pulling a same-colored rope to release a ball down a ramp. For the two individual devices, each participant could retrieve a ball by pulling the rope alone. Both ends of the rope were accessible on the collaboration device so that participants needed to pull together to retrieve the rewards. On both types of devices, the ends of each rope were wound around a wooden stick in front of the devices (height = 16 cm). We used two types of balls (diameter = 3 cm) as rewards in the study: High-rewarding balls were colorful (red or blue), rattling plastic balls placed inside a transparent and shiny round cover. Low-rewarding balls were plain wooden balls with similar colors. Participants could drop the rewards into different types of containers. Children could place the low-rewarding balls in a green plastic box (10 cm x 6 cm x 4 cm) and put the high-rewarding balls into the snout of a golden-colored toy elephant, producing attractive jingle sounds. We used a white curtain (70 cm x 70 cm), placed centrally in front of the devices, as a visual barrier between participants (see Fig 1A for an illustration of the collaboration task).

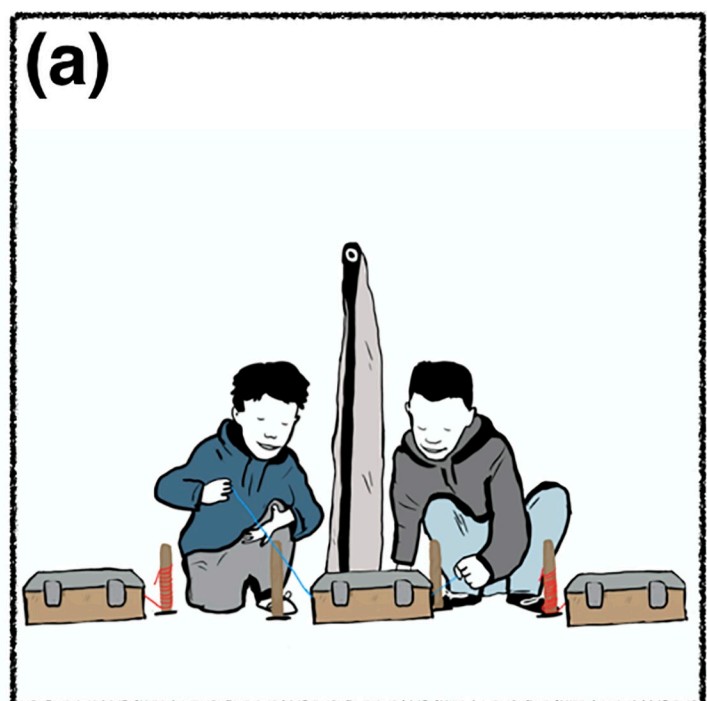 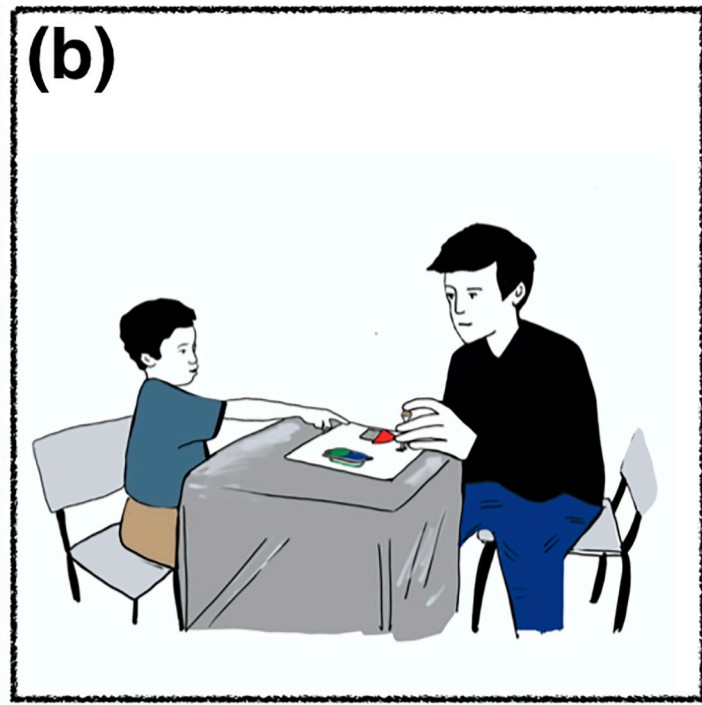

**Fig 1. Experimental set-up.** (a) Collaboration task; (b) Theory of Mind task.

**Theory of Mind.** We used an adaption of a five-point Theory of Mind scale [52] to assess children's social-cognitive abilities. This scale includes tasks on the concepts of Diverse Desires, Knowledge Access, Contents False Beliefs, Diverse Beliefs, and Hidden Emotions. We also tested children with an additional task on Explicit False Belief. However, due to an experimental error in one population, we did not include data on Explicit False Belief in our statistical analyses but used the original five-point scale instead.

We adapted the original scale as outlined below to meet the requirements of cross-cultural research. All modifications closely mirrored the original equivalent. Only minor changes in stimulus material were implemented to guarantee cultural appropriateness and familiarity. We used toy figurines with common appearance and names for each population. In the task on Diverse Desires, we implemented images of nuts and candy. For the task on Knowledge Access, we used a matchbox as a container and a toy cat as further stimulus material. We further utilized an empty fish tin as a container together with a cow figurine to test Contents False Belief. To assess Diverse Beliefs, we used comic images of a green bush and a wooden hut. We assessed Hidden Emotions using the original scale [52] as well as drawings of the back view of a child (see Fig 1B for an illustration of the experimental set-up during the assessment of Theory of Mind).

## Procedure

**Collaboration task: Training trials.** Participants entered the quiet study room with an adult male experimenter and sat down on the cushion next to a laptop from which instructions were given. Children were told that they could obtain rewards from the devices. Next, they saw how low-rewarding balls could be dropped into the respective container. Each child received a low-rewarding ball to put it into the container herself. The same procedure was then repeated

with the high-rewarding balls. In the next video sequence, an adult model appeared next to the left individual device and obtained a low-rewarding ball by unraveling and pulling the rope individually. A similar video started in which a second adult model retrieved a low-rewarding ball from the right individual device. Each child could practice the retrieval of balls from the two devices themselves. Subsequently, children saw both models on the laptop screen acting in the collaboration condition. Each model unraveled the rope before both pulled the rope collaboratively. Again, children could retrieve the balls from the collaboration device themselves.

**Collaboration task: Test trials.** Following training trials, dyads engaged in eight test trials. The first four test trials consisted of each a combination of reward (high vs. low) and condition (individual vs. collaboration) presented in a counterbalanced order. The order of conditions was then repeated for test trials five to eight. Thus, each child participated in each scenario twice during the test phase. Every trial was introduced by a video in which children were informed about the next round. Participants then saw a video sequence showing a model retrieving the ball either alone or collaboratively, depending on the condition.

Meanwhile, the experimenter baited the devices allowing children to be confronted with a similar situation to that previously seen on the laptop screen. Besides baiting the devices, the experimenter kept interactions with participants at a minimum. During individual trials, only one individual device was baited at a time, and participants pulled one after another. Here, the left device was always baited first to avoid confusion in participants.

**Collaboration task: Forced-choice trial.** After eight test trials, the experimenter announced a final forced-choice trial. The experimenter baited all devices with high-reward balls. Children decided on their own whether to retrieve a ball from the collaborative or the individual device. Each action resulted in a similar outcome, and children were free to talk to each other to coordinate their behaviors. Following this trial, children received candy (Skittles) for their participation (see Fig 2 for illustrations of the procedural steps).

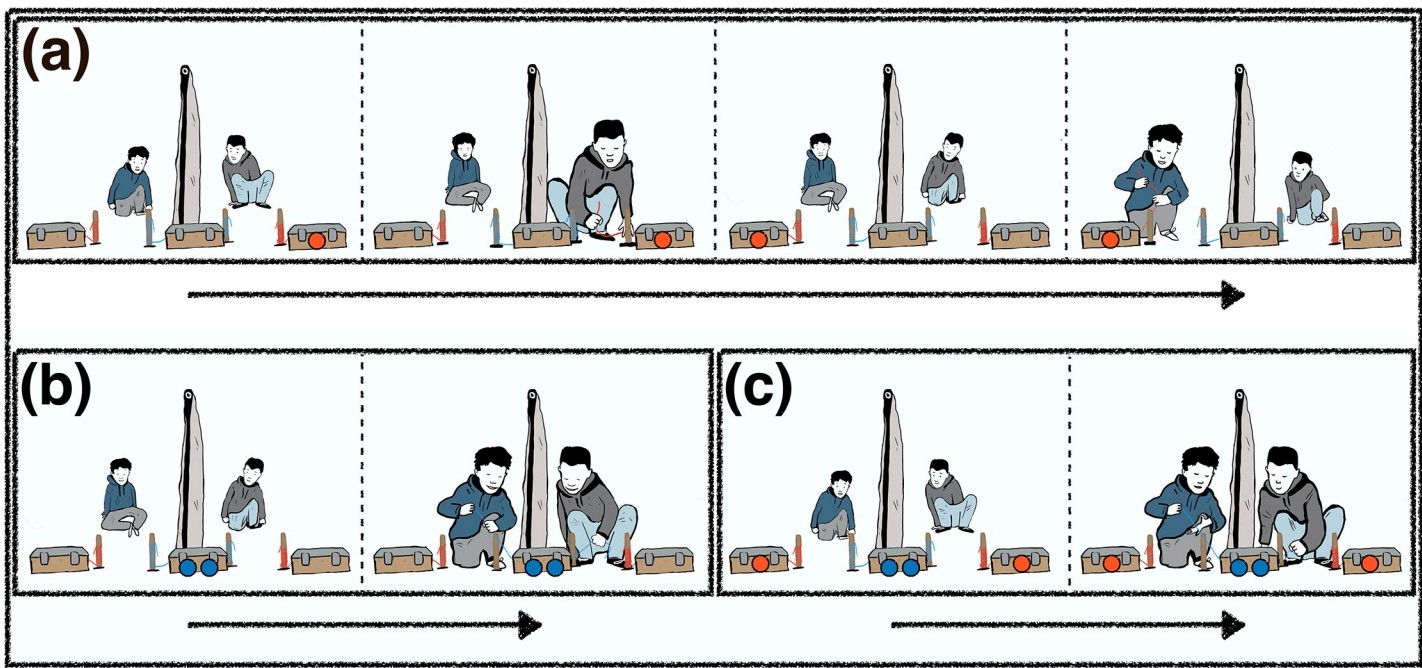

**Fig 2. Procedural steps.** (a) Individual and (b) collaboration trials; at the beginning of each trial children sit next to each other facing the devices; depending on condition, one (individual) or both (collaboration) participants then move toward the devices and pull the rope either alone or together with the peer; c) forced-choice trial with each device baited.

**Theory of Mind.**    Task manipulation and coding instructions were similar to the original test battery. Tasks were introduced to children in a fixed order (Diverse Desires, Knowledge Access, Contents False Beliefs, Diverse Beliefs, Hidden Emotions). Children were tested individually in a separate test session. Children were sitting on a chair next to the experimenter. Study materials were presented on a table between child and experimenter. Tests on Theory of Mind and social motivation were conducted within a maximal delay of 30 days for each child.

## Coding

**Social motivation: Emotional expressions during the collaboration task.**    We recorded children's behavior using a camcorder placed approximately 1.5 m in front of the devices (see Figs 1A and 2). For coding children's emotional expressions, we cut videos for each test trial separately. Each such sequence began when children started moving to the device, and the video sequence lasted until children picked up their respective ball. One video sequence lasted approximately ten seconds and was coded without sound. The first author coded each video with regards to children's happiness in the situation on a scale from 0 (neutral expression) to 4 (very happy expression). Using similar approaches, previous research reports good to excellent interrater-reliability between raters from different populations [53], since display rules of happiness are performed and recognized robustly and reliably across cultural contexts [54, 55]. Such ratings from tape are also reported to correlate almost perfectly with those of well-established measurements for affect, such as Baby FACS [56]. Initially, we also implemented a code of -1 (negative expression) in our coding scheme but did not make use of this because participants did not express negative emotions during test trials. Statistical analyses and data illustrations are thus given in the range from 0 to 4.

**Social motivation: Preferential choices during the collaboration task.**    Finally, the first author coded children's behavior on the forced-choice trial as a categorical variable. When deciding to pull individually, children received a score of 0. When choosing the collaboration device, children received a score of 1. If children tried to pull both ropes simultaneously, we scored them as 0.5.

A second German coder, blind to hypotheses and condition, coded a randomly selected quarter of participants per population. Interrater reliability for the affect data was high ($ICC = .73$). Regarding forced-choice data, reliability was excellent ($\kappa = .94$).

**Theory of Mind.**    To code children's Theory of Mind, we transcribed children's behaviors and verbal utterances. From these transcriptions, we rated each task one by one according to the coding instructions of the original task battery [52]. That is, children's correct answers were scored as 1, whereas incorrect answers were scored as 0. We then calculated a Theory of Mind score by adding all correct answers in the tasks on Diverse Desires, Knowledge Access, Diverse Belief, Contents False Belief, and Hidden Emotions. Children could thus gain a Theory of Mind score between 0 and 5. A second coder rated all transcriptions. Interrater agreements were excellent both on task level ($\kappa_{Diverse\ Desires} = .95$; $\kappa_{Knowledge\ Access} = .95$; $\kappa_{Diverse\ Beliefs} = .97$; $\kappa_{Contents\ False\ Belief} = .90$; $\kappa_{Hidden\ Emotions} = .96$) and with regards to overall Theory of Mind score ($ICC = .98$).

## Statistical analyses

We analyzed the data in *R* [57] and our statistical analyses can be grouped into three separate parts. To analyze children's social motivation across populations, we ran models on children's emotional expressions (Model 1) and their forced-choice decisions (Model 2). We further investigated whether children's Theory of Mind abilities could be predicted by children's emotional expressions and their forced-choice decisions (Model 3). As a first step, we fit the models using the package *lme4* [58] and tested the overall impact of the predictors in order to avoid

multiple testing issues [59]. To this end, we ran likelihood ratio tests comparing a full model comprising of the predictors of interest as well as control variables with a null model lacking those predictors. If this analysis indicated statistically significant effects of the predictors on the outcome, step two in the analyses covered further likelihood ratio tests comparing the full model with reduced models in which the respective predictors were excluded one by one. Each effect is described by reporting means and standard deviations for each level of the categorical predictors or by naming model estimates and standard errors for metric predictors. During this step, we tested for two-way interactions between predictors but excluded interaction terms if they did not reach statistical significance (detailed results of these analyses are given in the S1 File). All statistical tests were two-sided. All assumption checks, including variance-inflation-factor, normality distribution of residuals (for the linear models), and overdispersion, revealed no issues for any of the three models.

**Social motivation: Emotional expressions during collaboration task.** We investigated whether participants expressed different degrees of positive affect after retrieving the balls depending on population, condition, reward, and age (z-standardized) by calculating a linear mixed model with Gaussian error distribution. We averaged both trials of each combination (high reward vs. low reward) and condition (collaboration vs. individual) as a dependent variable to get a reliable indicator of children's emotional expressions across experimental manipulations. We also included sex as a control variable. A prior analysis showed that trial (1–4; standardized), position of the child (left vs. right), and color of collaboration device (blue vs. red) were marginally related to children's affect ($\chi^2_{\text{Trial}} (1) = 3.67$, $p = .055$; $\chi^2_{\text{Position*Population}} (2) = 9.53$, $p = .008$; $\chi^2_{\text{Color}} (1) = 3.09$, $p = .079$). Thus, we included these control variables in the statistical model. We further included the random effects of condition, reward, and trial for both dyads as well as each subject in the model. Trial was further included as a random intercept.

**Social motivation: Preferential choices during collaboration task.** Here, we focused on whether children's behavior during forced-choice trials was influenced by population and age. We ran a generalized linear mixed model with binomial error distribution. Before running the model, we excluded those children who did not show a preference for either individual activity or collaboration on this trial by pulling both ropes simultaneously. The frequency of these choices did not differ between populations ($n_{\text{German}} = 6$; $n_{\text{Hai||om}} = 6$; $n_{\text{Ovambo}} = 4$). We further included sex as a control variable. Finally, dyad was included as a random intercept. To test whether children's choices differed from chance level, we ran binomial tests for each group separately.

**Theory of Mind.** Running a linear mixed model, we investigated whether children's Theory of Mind was predicted by their emotional expressions depending on condition and their forced-choice behaviors in the pulling game. For this purpose, we calculated an affect index by subtracting participants' mean affect scores across individual trials (high-reward & low-reward) from the mean scores of the collaboration trials. An index of 0 thus indicated that children's positive expressed emotions did not vary between conditions. Positive indexes indicated more positive emotions expressed during collaboration. Negative indexes indicated a more positive affect when pulling individually. The dependent variable was children's Theory of Mind score. Forced-choice decisions and affect index were included as predictors. Further, populations, age (standardized), and sex were included as controls. To account for the dyadic situation in which the predictors were assessed, dyad was included as a random intercept.

## Results

### Social motivation: Emotional expressions during the collaboration task

The comparison between a full model and a null model indicated a strong effect of the predictors on the outcome ($\chi^2 (14) = 96.32$, $p < .001$). As a next step, we then assessed the statistical

significance of the 2-way interactions between the predictors. This analysis did not indicate an interaction between reward and the other predictors. In this model, the effects of the interactions between population and condition as well as between condition and age were statistically significant, which is why we ran a reduced model comprising the 2-way interactions between population, age, and condition, the main effect of reward, and the control variables.

Children across populations showed higher positive affect during collaboration, as compared to individual trials. However, this size of this effect differed between societies (Populations*Condition: $\chi^2$ (2) = 31.62, $p$ < .001; see Fig 3). German children showed the strongest bias for collaboration ($M_{Collaboration}$ = 1.68, $SD_{Collaboration}$ = 0.76; $M_{Individual}$ = 1.21, $SD_{Individual}$ = 0.78), followed by Ovambo children ($M_{Collaboration}$ = 1.49, $SD_{Collaboration}$ = 1.00; $M_{Individual}$ = 1.28, $SD_{Individual}$ = 0.99). Among Hai‖om children, this tendency was markedly smaller ($M_{Collaboration}$ = 1.37, $SD_{Collaboration}$ = 1.09; $M_{Individual}$ = 1.31, $SD_{Individual}$ = 1.04).

With age, children became more likely to show positive emotional expressions in collaborative as compared to individual trials (Condition*Age: $\chi^2$ (1) = 3.85, $p$ = .050). Furthermore, children expressed more positive emotions in trials with high rewards ($M$ = 1.44, $SD$ = 0.97) as compared to trials with low rewards ($M$ = 1.34, $SD$ = 0.95; $\chi^2$ (1) = 15.03, $p$ < .001). Additional analyses of the effects of the predictors separately for each population is given in the S1 File.

## Social motivation: Preferential choices during the collaboration task

Comparisons between full and null model showed a strong effect of the predictors population and age on the children's preferential choices ($\chi^2$ (5) = 26.34, $p$ < .001). That is, the ontogenetic trajectories of children's preference for collaboration over individual activity varied across the three populations (Population*Age: $\chi^2$ (2) = 7.29, $p$ = .026). Only German children showed a preference for collaboration from ages 3 to 8 ($p$ = .007). Hai‖om children, in contrast, preferred the individual option over collaboration ($p$ < .001). Among both of these populations, these tendencies appeared to consolidate with age (see Fig 4). Overall, Ovambo children showed no preference for either option ($p$ = .567). However, while younger Ovambo children preferred collaborative activities, this tendency reversed among older children as those preferably chose the individual option.

To further substantiate the construct validity of both proxies for social motivation in the current study, we also ran an additional exploratory analysis in which we added children's affect index as a predictor of children's preferred choices to the model. This analysis revealed that the link between children's affect index and their subsequent preferences varied with age (Affect Index*Age: $\chi^2$ (1) = 4.83, $p$ = .028). To further elaborate on this interaction, we visualized it by plotting the effect of affect index on children's preferential choices for three distinctive subsamples consisting of children younger than 5.5 years of age ($n_{Young}$ = 92), those between 5.5 and 6.5 years of age ($n_{Intermediate}$ = 56), and those older than 6.5 years ($n_{Old}$ = 92; see Fig 5). From these plots, it appears that the link between both proxies for social motivation is apparent among younger children who are more likely to prefer collaboration if they have expressed more positive emotions during previous collaboration trials. With increasing age, however, the two proxies for social motivation become dissociated. Notably, including children's affect index into the model did not alter the pattern of results described above, as the interaction between population and age remained statistically significant (Population*Age: $\chi^2$ (2) = 9.07, $p$ = .011).

## Theory of Mind

Full-null model comparisons in step 1 suggested that both proxies for children's social motivation had an effect on children's Theory of Mind performance ($\chi^2$ (3) = 10.53, $p$ = .015). Initial

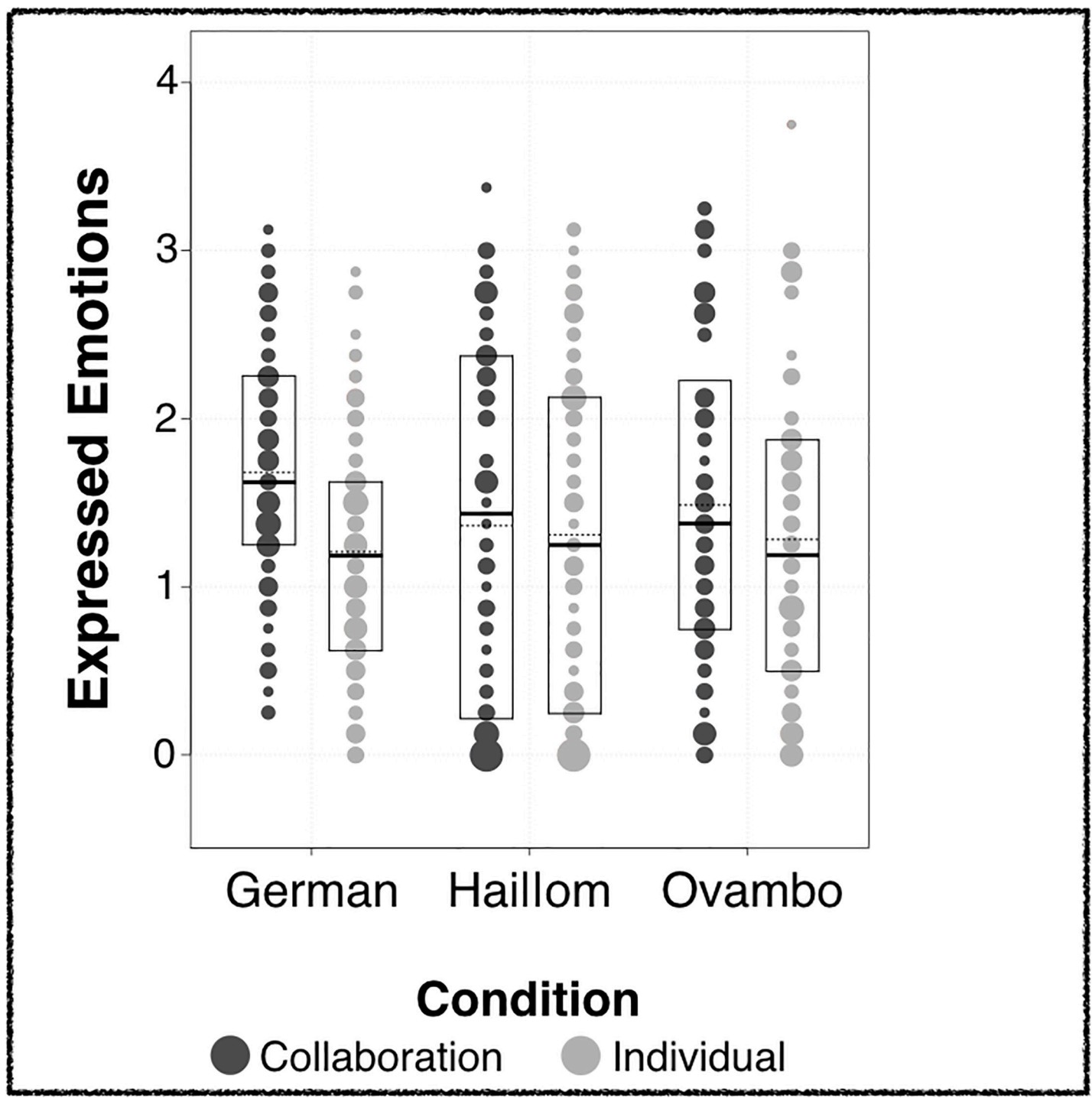

**Fig 3. Expressed emotions.** Positive emotions displayed across conditions; bubble sizes are proportional to data points; boxes indicate quartiles; bold horizontal lines indicate group medians; dotted horizontal lines indicate group means.

analyses did not indicate an interaction between children's affect indexes and their preferential choices ($\chi^2$ (1) = 0.40, $p$ = .526). As such, main effects of both predictors are reported.

Across populations, children's Theory of Mind performance could be predicted by their positive emotional expressions selectively during collaboration ($\chi^2$ (1) = 7.61, $p$ = .006). That

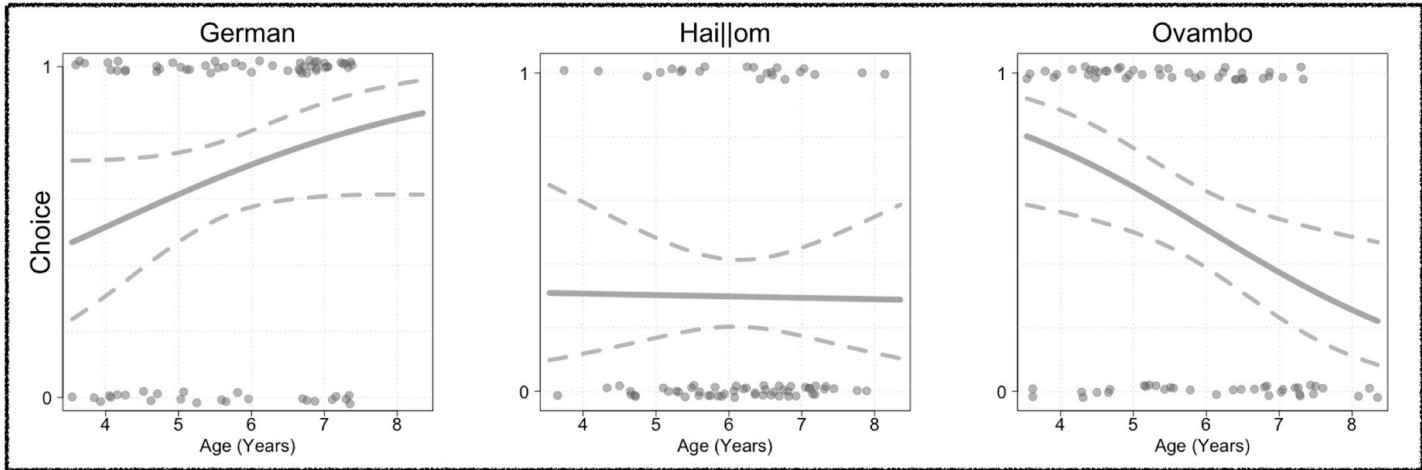

**Fig 4. Preferential choices.** Effect of age on choices (fitted values of Generalized Linear Models containing the same Predictors and Controls as the original model are plotted (bold lines) together with 95%-CIs (dotted lines); data points are jittered on the y-axis for visualization purposes).

is, children who selectively expressed more positive emotions during collaboration, as compared to individual activity, scored higher on the Theory of Mind scale (Estimate ± SE: 0.41 ± 0.15, see Fig 3). In contrast, children who chose collaboration during forced-choice trials showed only marginally higher Theory of Mind skills than those who preferred the individual option ($M_{Collaboration}$ = 2.39, $SD_{Collaboration}$ = 1.51; $M_{Individual}$ = 2.03, $SD_{Individual}$ = 1.34; $\chi^2$ (1) = 3.02, $p$ = .082). We also observed an effect of age on children's Theory of Mind such that older children from showed improved Theory of Mind skills across populations (0.46 ± 0.07; $\chi^2$ (1) = 37.62, $p$ < .001). Further, Theory of Mind skills differed across populations ($\chi^2$ (2) = 99.08, $p$ < .001). German children ($M_{German}$ = 3.43, $SD_{German}$ = 1.22) scored higher than Hai‖om children ($M_{Hai‖om}$ = 2.03, $SD_{Hai‖om}$ = 1.06). Hai‖om children scored higher than Ovambo children ($M_{Ovambo}$ = 1.15, $SD_{Ovambo}$ = 0.95). The effects of liking on Theory of Mind are illustrated in Fig 6.

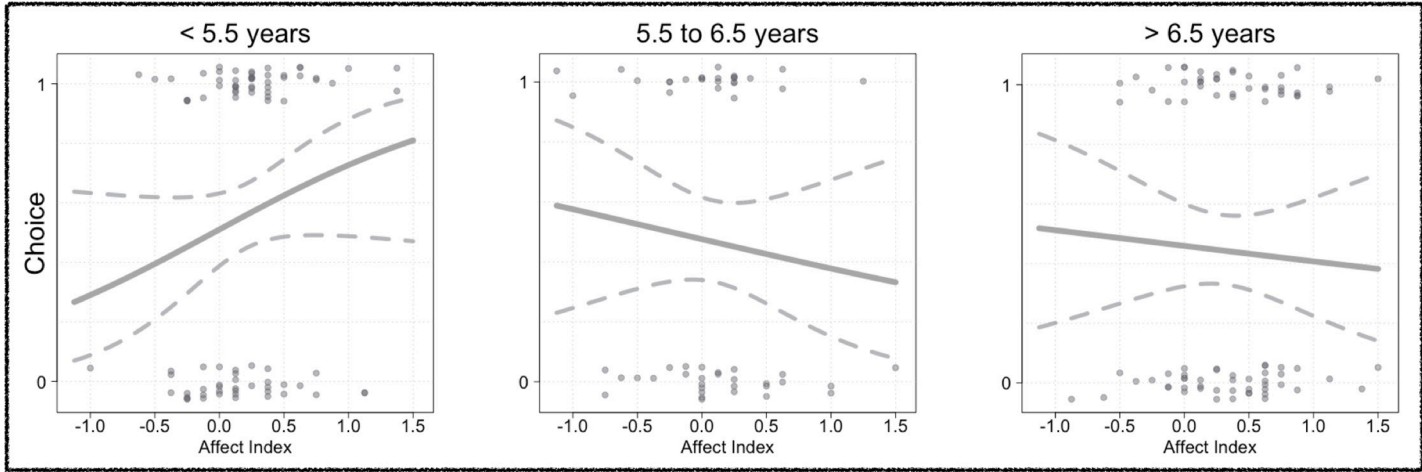

**Fig 5. Preferential choices.** Effect of positive expressed emotions selectively during collaboration trials on preferential choices across populations; affect indexes > 1 indicate that children express more positive emotions during collaboration, as compared to individual trials (fitted values of Generalized Linear Models containing the same Predictors and Controls as the original model are plotted (bold lines) together with 95%-CIs (dotted lines); data points are jittered on the y-axis for visualization purposes).

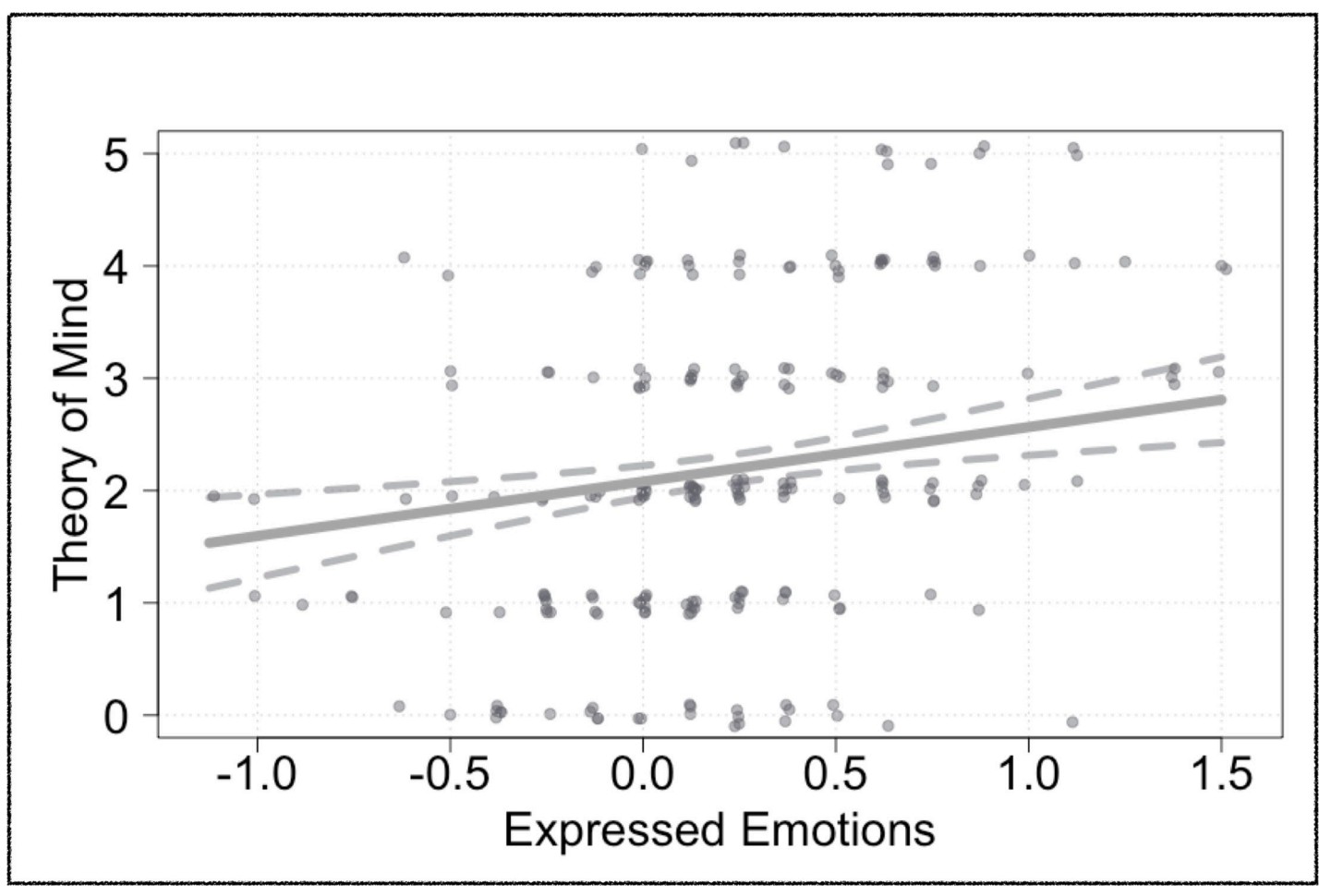

**Fig 6. Theory of Mind.** Effect of expressed emotions selectively during collaboration (affect index) on Theory of Mind scores; affect indexes > 1 indicate that children express more positive emotions during collaboration, as compared to individual trials (fitted values of Generalized Linear Models containing the same Predictors and Controls as the original model are plotted (bold lines) together with 95%-CIs (dotted lines); data points are jittered on the x- and y-axis for better identification of the individual data).

## Discussion

Our results reveal both cross-cultural continuity and variability in children's social motivation and its link to social cognition. Across populations, children expressed more positive emotions when obtaining a reward collaboratively as compared to doing so individually. This effect was most substantial among German, attenuated among Ovambo, and weakest among Hai‖om children. Children's explicit choices of collaboration varied more strongly across populations: German children preferred dyadic peer collaboration over individual activity. Only younger Ovambo children preferred to pull collaboratively, whereas older participants preferred individual activities. Across the age range tested, Hai‖om children preferred the solitary activity. Independent of population and age, children's positive emotional expressions selectively during collaboration predicted their Theory of Mind skills at an individual level. Together these results indicate that the positive emotions accompanying dyadic peer collaboration are cross-culturally recurrent and ontogenetically linked to children's emerging social-cognitive skills.

The current results lend partial support to universalists' claims regarding the role of social motivation in early ontogeny [2, 17]. Accordingly, children express positive emotions during

collaborative social interactions. These emotions, in turn, can be at the base of a multitude of species-typical levels of (ultra-)sociality [60]. For example, experiencing positive emotions through collaboration may boost children's prosocial behaviors by motivating them to join and help in adults' activities [61], or to build positive reputations with potential collaborators [62]. The positive emotions experienced accompanying collaborative interactions might facilitate the ontogeny of Theory of Mind, which may, in turn, boost the reward value of social interactions by enabling effective behavioral coordination between partners.

In contrast to children's emotional expressions, their explicit choice of collaboration varied starkly across populations. These cultural differences consolidated with age, resonating with previous work documenting that culturally-specific social behaviors emerge over middle childhood [63–65]. Around this age, cultural practices give rise to normative obligations and external rewards resulting from collaborative activities. The stable preference for collaboration among German children may be a cumulative result of Western pedagogy and praise for collaborative efforts, underpinned by a species-typical proclivity to collaborate. The preference for individual activity among older Ovambo and Hai‖om children may, in contrast, be best explained by the impact of parental cultural models focusing on autonomy, possibly attenuating children's intrinsic motivations to collaborate: The Hai‖om emphasize child autonomy as an essential socialization goal from early on in ontogeny [34]. Given a choice between collaboration or individual work toward the same goal may have led Hai‖om children to prefer individual action to avoid being dependent on the collaborative partner [35]. Among the Ovambo, younger children displayed an initial preference for collaboration, possibly driven by children's proclivity to collaborate, which may be additionally fostered by a cultural focus on interpersonal relatedness ([31, 32], see also [40]). However, with increasing age, children preferred the individual option when given a choice. Ovambo children are increasingly requested to autonomously contribute to household chores and subsistence activities as they grow up. This action autonomy marks a central socialization goal among many traditional farming populations, such as the Ovambo [29, 40]. It might be that older Ovambo children chose individual activities to demonstrate their action autonomy.

In line with these findings, prior studies on children's social learning preferences have documented similar ontogenetic shifts towards autonomy across rural, non-Western societies. For example, the influence of a majority on children's social learning declines over middle childhood, which may be driven by increasing egocentrism around this age [65]. It remains unclear, however, to which degree the developmental trends in this study are specific to the context of collaboration or whether they reflect motivational shifts in social interactions per se.

More generally, we argue that the developmental trajectories observed in the current study highlight the necessity to conceptualize social motivation and cognition within both a cross-cultural *and* a developmental perspective. For example, the developmental continuity in German children's preferences for collaboration suggests that even such continuities may result from a dynamic process involving developmental shifts interacting with culturally-specific variables.

Another possible explanation for both Ovambo and Hai‖om children's lower preference for collaboration when compared to their German counterparts may lie in the dyadic setting in which their behaviors were assessed. While such interactions are actively encouraged within Western pedagogy [14], children from non-Western societies may be much more used to social interactions within groups consisting of three or more individuals [10, 14, 16]. In a recent study, Keller and colleagues assessed dyads and triads of young children from urban Germany in their tendency to solve challenging tasks cooperatively [15]. They found these children to do so primarily within dyadic settings, whereas triadic cooperation was observed at much lower rates. In other studies, children from non-Western societies have shown strong

capacities for collaboration within triadic settings and beyond [10]. Whether the current findings regarding children's social motivation for peer collaboration can be generalized beyond dyadic settings warrants further investigation.

We applied a conservative coding scheme to rate children's preferences during forced-choice trials. That is, children's choices were only coded as *collaboration* if they approached the collaboration device. We followed this conservative approach based on previous work on children's collaborative decision making (e.g., [66]). This approach is in line with the current definition of collaboration as a joint endeavor in which two or more individuals engage to pursue a common goal. However, children's joint decisions to pull individually may also reflect a social commitment. In eight of the 40 German dyads both children pursued the individual option, in contrast to only 17 Hai||om dyads and 14 Ovambo dyads. At the same time, 19 German dyads and 15 Ovambo dyads, but only four Hai||om dyads decided to jointly pursue the collaboration device. Taken together, children agreed on one or the other option in most trials, even though Hai||om showed greater autonomy in their preferences within dyads. Again, this pattern may reflect cultural schemas regarding autonomy and social interdependence which are often reported for foraging societies, such as the Hai||om (e.g., [36]).

We found that individual differences in children's emotional expressions during dyadic collaboration predicted their Theory of Mind skills. We propose that positive emotions experienced during collaboration increase the reward value of such activities. As a consequence, the quality and frequency with which children engage in collaborative interactions may increase. Since collaborative activities provide a particularly valuable learning context to foster Theory of Mind development by demanding mental coordination and perspective-taking [3, 17], this increase provides children with essential opportunities for socio-cognitive development. A second (reverse) interpretation could be that children's emerging Theory of Mind enables successful coordination with others during collaboration, which, in turn, might make collaborative activities more rewarding. We argue that a combination of both processes is most plausible: Social motivation and social cognition build on one another through the experience children gain in social activities, such as peer collaboration. Again, future studies will need to investigate whether this link is specific to collaborative activities or whether it applies to social interactions in general [2, 39].

We assessed children's Theory of Mind in order to test whether social motivation would predict children's Theory of Mind across diverse populations. It has to be noted that the current study design does not allow us to draw definite conclusions with regard to the causality of this link. The study was designed to test predictions put forward in the *Social Motivation Theory of Autism* [2] across diverse populations. In this framework, social motivation is conceived as an ontogenetic driver of young children's social-cognitive development. In support of this notion, Burnside and colleagues [42] found young children's social orientation, another proxy of social motivation, to relate to their performance on an (implicit) false belief task. While the current study thus yields initial support that such a link between social motivation and Theory of Mind may indeed be a cross-cultural regularity, longitudinal research is highly needed to address the causality of this proposed relation.

It is important to note that we did not have specific predictions regarding cross-cultural differences or homogeneity in Theory of Mind. The current results, however, add to a growing body of research documenting substantial variation in children's (explicit) Theory of Mind across populations [43–45]. It is most likely that a combination of different factors may underlie the cross-cultural variability observed here: To achieve high scores on the current set of verbal Theory of Mind tasks, German children may benefit from the more frequent conversations about mental states they experience in their daily lives [38]. One practice that may be particularly influential in this regard is parental mind-mindedness, or their "proclivity to view their

children as mental agents" ([64], p. 1297; see also [65]). In one study, Hughes and colleagues [67] assessed parental mind-mindedness in adults from Hong Kong and the U.K. in addition to their children's Theory of Mind skills. While these researchers documented cultural variation in either variable, both phenomena were linked within populations. It thus appears that parenting practices may underlie developmental variation in Theory of Mind across populations, while also relating to interindividual variation within populations.

Another candidate variable to underlie the observed variation in Theory of Mind performance is German children's exposure to dyadic, adult-child pedagogy, which may have privileged these children as compared to their Namibian counterparts. Finally, the focus on authoritarian parenting practices among the Ovambo may have led to lower scores in the Theory of Mind tasks than both German and Hai‖om children (see also [51, 68]). Accordingly, authoritarian parenting practices may encourage children to focus primarily on the behavioral outcomes of their actions and whether they comply with the demands implied by caregivers. Holding and comparing multiple perspectives in parallel, which is an essential component of Theory of Mind reasoning, may be of lower relevance in populations in which hierarchical relatedness and obedience are prioritized over individual autonomy. While we thus speculate that a combination of these factors may account for the variation in children's Theory of Mind performance observed here, we highlight that the current results need to be interpreted with caution. Given the current focus, a targeted analysis of cultural variation in children's Theory of Mind skills is beyond the scope of this study. Future studies will need to target how children's Theory of Mind is embedded in their daily lives (e.g., [69, 70]) in order to draw valid conclusions on developmental variation between population.

It has to be noted that the current study does not contradict previous findings on the collaborative and coordinative skills of young children from various traditional, non-Western populations [6–8, 10]. Given that children in the current study were unanimously capable of collaborating with their peers when they had to, these findings should not be understood as indicating cultural differences in children's skills for collaborative problem-solving or collaborative skills per se. In the present study, children across all ages and populations mastered the collaboration game flexibly and skillfully. How children's social motivation for peer collaboration relates to their coordinative skills to collaborate in more complex and challenging situations than the ones presented here deserves further inquiry (e.g., [10, 16]). For example, peer collaboration between multiple individuals may insinuate higher demands on social coordination than dyadic interactions [10, 15]. Further, collaboration which is embedded in subsistence activities and household chores may comprise different affordances than collaboration within playful settings (such as in the current study). Work by Rogoff and colleagues [27, 26, 37] has emphasized activities for young children in Western, industrialized societies are often framed as playful and non-functional in the context of subsistence activities. If children are asked to contribute within adult activities this may alter the reward value of collaboration notably.

The current study also revealed an interesting pattern regarding the construct validity of both proxies for social motivation investigated here: Younger children's positive expressed emotions selectively during collaboration trials predicted their subsequent preferences. That is, these children were more likely to pick the collaborative option if they had displayed more positive emotions during previous trials. This pattern was not evident among older children, where both proxies for social motivation were dissected. Since our study was motivated by the *Social Motivation Theory of Autism* [2], both proxies for social motivation were implemented to fully capture the reward value of social interactions. In accordance with this research [2, 71, 72], we focused both on children's *liking* of collaboration (e.g., their positive expressed emotions) and their *seeking* of such (e.g., their preferential choices). Interestingly, the scientific literature on rewards emphasizes that seeking and liking reflect dimensions that complement

each other, but which are not necessarily linked on an individual level [71]. In light of the current study, it is not surprising that children did not necessarily seek what they liked. While younger children may choose their preferred option according to what they like, older children may encounter a more complex decision. At this age, societal norms, such as those structuring peer collaboration and action autonomy, become increasingly relevant for young children (e.g., [73]). Depending on populations, some children may have thus sought the individual option to act autonomously and without peer support, whereas others may have preferred to collaborate instead to fulfill normative expectations regarding their willingness to engage with peers in dyadic collaboration.

To conclude, we found that children's motivational preference for engaging in dyadic collaboration over individual action varies across cultures and ontogeny. At the same time, we found children from three diverse populations to express more positive emotions during dyadic peer collaboration, as compared to acting individually. This cross-culturally recurrent aspect of social motivation predicted children's Theory of Mind in all three populations. We propose that an enhanced positive emotionality in response to collaboration increases the quality and quantity of children's early social interactions, boosting learning opportunities for social-cognitive skills across diverse cultural contexts.

## Supporting information

**S1 File.**
(DOCX)

**S2 File.**
(DOCX)

**S1 Data.**
(CSV)

## Acknowledgments

We thank D. Tjizao and L. Hupfer for their help in conducting the study. We thank R. Mundry for statistical advice. We further thank T. Toppe, M. Thiele, and two anonymous reviewers for helpful comments on previous versions of the manuscript, and all children, parents, and institutions for their immense cooperation.

## Author Contributions

**Conceptualization:** Roman Stengelin, Robert Hepach, Daniel B. M. Haun.

**Data curation:** Roman Stengelin.

**Formal analysis:** Roman Stengelin, Robert Hepach.

**Funding acquisition:** Daniel B. M. Haun.

**Investigation:** Roman Stengelin.

**Methodology:** Roman Stengelin, Robert Hepach, Daniel B. M. Haun.

**Project administration:** Roman Stengelin.

**Supervision:** Robert Hepach, Daniel B. M. Haun.

**Visualization:** Roman Stengelin.

**Writing – original draft:** Roman Stengelin.

**Writing – review & editing:** Robert Hepach, Daniel B. M. Haun.

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
