## [Decision Letter · Decision Letter 0]

18 Jun 2020

PONE-D-20-07321

Cultural Variation in Young Children’s Social Motivation for Peer Collaboration and its Relation to the Ontogeny of Theory of Mind

PLOS ONE

Dear Dr. Stengelin,

Thank you for submitting your manuscript to PLOS ONE. After careful consideration, we feel that it has merit but does not fully meet PLOS ONE’s publication criteria as it currently stands. Therefore, we invite you to submit a revised version of the manuscript that addresses the points raised during the review process.

We look forward to receiving your revised manuscript.

Kind regards,

Valerio Capraro

Academic Editor

PLOS ONE

Journal Requirements:

2. During our internal checks, the in-house editorial staff noted that you conducted research or obtained samples in another country. Please check the relevant national regulations and laws applying to foreign researchers and state whether you obtained the required permits and approvals. Please address this in your ethics statement in both the manuscript and submission information.

4. We note that Figure(s) 1 & 2 in your submission contain copyrighted images. All PLOS content is published under the Creative Commons Attribution License (CC BY 4.0), which means that the manuscript, images, and Supporting Information files will be freely available online, and any third party is permitted to access, download, copy, distribute, and use these materials in any way, even commercially, with proper attribution. For more information, see our copyright guidelines: http://journals.plos.org/plosone/s/licenses-and-copyright.

1.    You may seek permission from the original copyright holder of Figure(s) 1 & 2 to publish the content specifically under the CC BY 4.0 license.

Additional Editor Comments (if provided):

I have now collected two reviews from two experts in the field. The reviewers are both positive but suggest several revisions to improve the manuscript. Therefore, I am glad to invite you to revise your manuscript for Plos One. Needless to say that all comments should be addressed. I am looking forward for the revision.

Reviewers' comments:

Reviewer's Responses to Questions

**Comments to the Author**

1. Is the manuscript technically sound, and do the data support the conclusions?

Reviewer #1: Partly

Reviewer #2: Partly

2. Has the statistical analysis been performed appropriately and rigorously? 

Reviewer #1: Yes

Reviewer #2: I Don't Know

3. Have the authors made all data underlying the findings in their manuscript fully available?

Reviewer #1: Yes

Reviewer #2: No

4. Is the manuscript presented in an intelligible fashion and written in standard English?

Reviewer #1: Yes

Reviewer #2: Yes

5. Review Comments to the Author

Reviewer #1: Significance

This study examines affect during and choices to engage in either individual or collaborative activity to complete a task. The participants are children in Germany and rural Namibia aged 3 to 8. I commend the authors for doing work with these populations.

The study finds somewhat similar patterns of affect (positive) in collaboration versus solitary activity but very different patterns in engagement with collaboration. German children are the most collaborative and positive, younger Ovambo children engaged in relatively more collaboration than older children, and Hai||om children engage in little collaboration. The authors interpret the former as a sign of a cultural universal, and the latter as cultural differences.

Larger issues

Literature review

Although I am not a cultural researcher, I had some concerns about how children’s collaboration and adult supervision were framed in the literature review. In particular, there was sometimes a framing that suggested that Western children were more collaborative and more relational which seems to upend well established findings in the field - and which may shape interpretation of the findings here.

Paragraph 2 page 3 notes a contrast in findings between references 6, 10, and 13 and 4, 5, and 11. However, studies by Rogoff and colleagues show more sophisticated collaboration (and perhaps even qualitatively different collaboration as in “a single organism with many limbs”) in non-Western children. They do not argue that Western do not collaborate as is examined in the present study.

On page 4, paragraph 2 (“in many traditional…”), I don’t believe the argument in Rogoff and colleagues work is that adults do no supervise children in tasks, but they that they do so in a very different way than in Western style teaching. Also, adults may expect children to watch them (e.g., children watching and learning from an elder) in many non-Western cultures.

On page 4 paragraph 3 (“In traditional hunter-gatherer…”), it is noted that “child-centred pedagogy… is rare… because of [Ovambo and Haillom] cultural emphasis on individual autonomy”. Given that the cultural dichotomy is often framed as relational vs. autonomous, it would almost seem as if this article is reversing the usual pattern, and framing Western families as relational and non-Western families as autonomous/individual. It might be useful to consider some work by Rogoff and colleagues (notably Andrew Coppens) and “mock” and “meaningful” participation. Western parents often lavish praise on acts that they consider cute, but do not actually expect children to contribute to the home in a meaningful way. In contrast, in non-Western and Indigenous cultures, children’s participation is meaningful, and important. A child who is trying to raise resources for the family cannot make mistakes. The Kärtner group in Münster may also have some work in this regard.

Overall, these issues make me somewhat concerned about the framing of the collaboration task here. German and Western children are expected to collaborate with peers at school and in games and activities. For these children, the collaboration here could be one such game. Children in hunter-gatherer societies are likely expected to engage in real-life tasks in their community. How do they interpret the task here? Is it “just a game” and therefore not very important and/or not bound by the same norms as typical activity? Without observations from these communities (e.g., how children interact) it is hard to tell. Perhaps the authors can provide this information in a revision. It would be important to learn if the children had any exposure to formal schooling and when they got this exposure (i.e., starting at what age). I would be cautious about drawing “cultural universal” conclusions here if the framework of meaning is very different. I appreciate the authors drawing on some resources in the discussion (e.g., page 19) but perhaps these can be brought in sooner and integrated more substantively. I think the authors’ work with the rural Nambian populations is very interesting, but I also think they owe readers a richer account of these cultures and communities (and also respecting the fact that German culture is also its own culture, not just the “default”).

Methods

Why were the ages chosen? There are major developmental transitions, notably a qualitative shift in Theory of Mind in the age range tested (e.g., False Belief). Could analysis include a look into these issues?

I thought using ToM as a dependent for expressed emotion (itself a proxy for social motivation) was interesting, but also surprising. Could the authors talk a little more about their idea here? Typically, ToM is including as an independent variable. I see the link in the introduction between ToM and social motivation, but in the age ranges here, where at least some of the ToM skills tested are already in place in many three year olds, the relation could go in the other direction.

Discussion

As noted, perhaps some of the resources in the conclusion can be brought up in the introduction.

Minor

Perhaps mention Namibia in the abstract.

Reviewer #2: I enjoyed reading this MS, which presents a thoughtful and nicely structured cross-cultural examination regarding the relations between a motivation for collaboration, actual collaboration, and ToM among 4-8-year-olds from three different populations. The theory is relatively rich and the procedure is well-designed. Nonetheless, I do wish to raise several concerns with regard to the analysis, and believe that addressing these issues will make it much easier to assess the merit of the study in its’ entirety.

1. Procedure

1.1. I have to say that setting rewards of different kinds (high/low) with different kinds of devices (individual/collaborative) is a very good design for disentangling a motivation for obtaining a “better prize” and a motivation to “do something together”.

1.2. The authors should mention whether children within a dyad were familiar with each other or not.

1.3. Given that children were allowed to talk prior to their choices (line 252), how would the authors interpret children’s “coordinated decision” to pick individual devices? This is important especially when both the individual and the collaborative devices provide the same material reward. In other words, I urge the authors to explain and speculate whether there is any meaningful difference between a mutual decision to go on separate paths (individual device) and a mutual decision to take one path (collaborative device). Adding more information about the nature and frequencies of children’s communication may help.

2. Analysis

2.1. Model 1 assess children’s motivation for collaboration, but it was a bit unclear what was exactly included in the model:

a. Were all the possible interactions between population, condition, reward and age included? The authors report that interactions were tested (line 313), but they should detail exactly what was tested and what was found per model (i.e., non-sig 4-way interaction, non-sig 3-ways, non-sig 2-ways, etc.).

b. Lines 333: following the previous comment, what was the “preliminary analysis” in which no interactions were found with “reward”?

c. Line 362: The authors report an interaction between population and condition, but do not report the results of statistical tests that were done per population (i.e., was the difference among the German children is significant?).

d. Line 369: The authors report that overall, children expressed more positive emotions when played with high-value rewards. It is important to know whether a M.E for “reward” was also present in each population separately (“manipulation check”).

2.2. Model 2 assess children’s actual preferences for the individual or collaborative devices.

a. Can the authors explain why the “affect index” (from model 1) was not included as a predictor in model 2? If “affect index” will be included, then the authors can show a direct link between “positive emotions” during “individualistic or collaborative” tasks and children’s actual choices to engage in individualistic or collaborative tasks. This could shed light on differences at the population level as well as the individual level, for example, even though Ovambo children showed no significant preference for individualistic or collaborative tasks, it is plausible that those with a positive “affect index” favored the collaborative device whereas those with a negative “affect index” favored individual device.

b. Along the same lines, how can the authors explain Hai||om children’s similar motivation for individual and collaborative devices (model 1, line 365) but their significant preference for actually choosing the individual device? (e.g., Did Hai||om children who are motivated to collaborate (i.e., high “affect index”) inhibited their motivation and chose the individual device (i.e., forced-choice) in order to “achieve autonomy”?).

c. Figure 4 is interesting. Still, it will be informative to present the actual data points across the age range (so we can get an impression of how many children were per age group), as well as the CIs for the regression lines. To achieve this, I think three separate graphs per population will be more adequate.

2.3. Model 3 assess children’s ToM scores on the basis of their “affect index” (model 1) and forced-choice (model 2).

a. I found the logic behind this model a bit strange, and recommend the authors to better explain it. Specifically, the most valuable thing to predict in my opinion would be children’s forced-choice on the basis of their “affect index” and ToM score. For example, if a given population promotes collaboration (via cultural norms, as the authors suggest), then ToM is not needed at all as collaboration can be also achieved 1) via mere compliance to norms (as was found in line 397), or 2) via explicit verbal coordination (see point 1.3 above), and 3) as also seems to be the case among young Ovambo children (who collaborate even with low ToM scores).

b. In this case, a M.E for population can strengthen the author’s point regarding cultural differences in collaboration that are independent of individual differences (e.g., ToM).

c. The authors should also expand on possible reasons for big differences in ToM scores across populations (line 401), especially when such differences were not found in other cultures (e.g., res 41 and 44). It could definitely be that the ToM task that was used here is not suited for Ovambo and Hai||om, but I believe this should be discussed nonetheless.

6. PLOS authors have the option to publish the peer review history of their article (what does this mean?). If published, this will include your full peer review and any attached files.

Reviewer #1: No

Reviewer #2: No

---

## [Author Response · Author response to Decision Letter 0]

28 Jul 2020

Dear Dr. Capraro:

Thank you for the opportunity to revise our manuscript PONE-D-20-07321. We are grateful for the Reviewers’ encouraging assessment of our manuscript:

“I commend the authors for doing work with these populations“ (Reviewer 1)

“I enjoyed reading this MS, which presents a thoughtful and nicely structured cross-cultural examination […]” (Reviewer 2)

In the revised version of the manuscript we now address the points raised by both Reviewers. We outline the revisions we made point by point and have boldfaced the specific points.

Reviewer 1 wrote:

(1)

“Although I am not a cultural researcher, I had some concerns about how children’s collaboration and adult supervision were framed in the literature review. In particular, there was sometimes a framing that suggested that Western children were more collaborative and more relational which seems to upend well established findings in the field - and which may shape interpretation of the findings here.”

We have added further information to clarify that children from all three populations tested here learn to collaborate in their first years of life. At the same time, we argue that the contexts in which children typically do so may differ across cultures. We fully agree with the reviewer that social relatedness is typically given a higher emphasis among rural, non-Western when compared to urban, Western communities (especially in small-scale farming communities, such as the Ovambo). However, this does not necessarily imply that children from Western populations do not collaborate. Here, peer collaboration may be viewed primarily as a source of joy, rather than a social obligation (Keller, 2007). Numerous studies report that children in such populations collaborate frequently and are motivated to do so given the choice (e.g., Rekers et al., 2011; Tomasello et al., 2005, Hamann et al., 2011), but our current understanding of children’s motivation to collaborate suffers from a strong bias toward urban, Western participants:

“In sum, children from diverse populations engage in peer collaboration, even though populations show striking and systematic differences in how social interactions are framed culturally. In Western industrialized societies, such as urban Germany, adult caregivers typically consider play-like activities as an essential experience for their children. Notably, such experience is often detached from social obligations and family chores (37). Here, children flexibly choose with whom they want to interact, which reflects the cultural emphasis on psychological autonomy over action autonomy (29,38). In consequence, social interactions can be entered and dissolved depending on whether they are experienced as personally rewarding or not. In traditional farming societies, such as the Ovambo of Namibia, a mixture of social relations and action autonomy frames children’s socialization experience. Among traditional foraging societies, such as the Hai||om of Namibia, individual autonomy is emphasized in both the psychological and the action domain (29,33). It is, however, unclear whether and how such variation in socialization goals affects children’s social motivation to engage in peer collaboration, given that the vast majority of studies in this area have been conducted in urban Western populations (11,17,20). While social motivation has been hypothesized to be an ontogenetic driver of social-cognitive development in Western societies (2,3,17,39), it remains unclear whether this link can be generalized across populations.” (p. 5)

(2)

“Paragraph 2 page 3 notes a contrast in findings between references 6, 10, and 13 and 4, 5, and 11. However, studies by Rogoff and colleagues show more sophisticated collaboration (and perhaps even qualitatively different collaboration as in “a single organism with many limbs”) in non-Western children. They do not argue that Western do not collaborate as is examined in the present study.”

We agree that the dichotomous descriptions in the initial submission did not fully capture the more gradual differences in peer collaboration across societies. We now give a more detailed outline of previous research in the manuscript. (see point (1) above)

(3)

“On page 4, paragraph 2 (“in many traditional…”), I don’t believe the argument in Rogoff and colleagues work is that adults do no supervise children in tasks, but they that they do so in a very different way than in Western style teaching. Also, adults may expect children to watch them (e.g., children watching and learning from an elder) in many non-Western cultures.”

We agree and have tempered the description of socialization practices in traditional farming communities.

“In many traditional farming communities, for example, parents expect their children to learn from them by observing them and through active participation in daily activities (25,26). Here, children mostly engage with peers, rather than adults, playfully, even though play is typically embedded into daily chores and practices (27). Compared to urban, Western populations, parents incentivize and reward children’s socially appropriate behaviors and contributions less actively. Instead, children rely more on their interest to navigate interactions with peers and adults to learn and participate. Here, caregivers often value hierarchical relatedness, obedience, and conformity as central socialization goals (28,29). The Ovambo, for example, a Namibian agro-pastoralist population, emphasize these values in their childrearing practices (30–32): Children’s relatedness to the social group is emphasized by caregivers’ use of directive and assertive communication strategies (30,31). Young children are frequently tasked with household duties and demanded to exercise these autonomously and without much adult supervision.” (p. 4) 

(4)

“On page 4 paragraph 3 (“In traditional hunter-gatherer…”), it is noted that “child-centred pedagogy… is rare… because of [Ovambo and Haillom] cultural emphasis on individual autonomy”. Given that the cultural dichotomy is often framed as relational vs. autonomous, it would almost seem as if this article is reversing the usual pattern, and framing Western families as relational and non-Western families as autonomous/individual. It might be useful to consider some work by Rogoff and colleagues (notably Andrew Coppens) and “mock” and “meaningful” participation. Western parents often lavish praise on acts that they consider cute, but do not actually expect children to contribute to the home in a meaningful way. In contrast, in non-Western and Indigenous cultures, children’s participation is meaningful, and important. A child who is trying to raise resources for the family cannot make mistakes. The Kärtner group in Münster may also have some work in this regard.”

We agree that this information could be presented more precisely. In accordance with ethnographic literature from foraging societies, we aim to make the point that direct pedagogy may be considered inappropriate among foraging societies because of the importance given to autonomy in such contexts. Among traditional farming societies, such as the Ovambo, the opposite is true: Here, relatedness is emphasized in over (psychological) autonomy. Regardless of these differences, both contexts give emphasize direct pedagogy to a lesser degree than Western societies. We have clarified this information in the current version of the manuscript. 

We believe that the framework provided by Coppens and colleagues (2016) on cultural paradigms regarding children’s participation fits well to the distinction between psychological autonomy and action autonomy put forward by Keller and Kaertner (2013). We have now integrated both perspectives in the manuscript. (see point (1) above)

(5)

“Overall, these issues make me somewhat concerned about the framing of the collaboration task here. German and Western children are expected to collaborate with peers at school and in games and activities. For these children, the collaboration here could be one such game. Children in hunter-gatherer societies are likely expected to engage in real-life tasks in their community. How do they interpret the task here? Is it “just a game” and therefore not very important and/or not bound by the same norms as typical activity? Without observations from these communities (e.g., how children interact) it is hard to tell. Perhaps the authors can provide this information in a revision. It would be important to learn if the children had any exposure to formal schooling and when they got this exposure (i.e., starting at what age). I would be cautious about drawing “cultural universal” conclusions here if the framework of meaning is very different. I appreciate the authors drawing on some resources in the discussion (e.g., page 19) but perhaps these can be brought in sooner and integrated more substantively. I think the authors’ work with the rural Nambian populations is very interesting, but I also think they owe readers a richer account of these cultures and communities (and also respecting the fact that German culture is also its own culture, not just the “default”).”

We greatly thank the reviewer for raising this interesting point. Indeed, we believe that the collaboration was perceived as a game-like activity for children in all populations. Children everywhere were highly motivated to participate in the study, suggesting a strong incentive to engage in the tasks that would not be predicted if the tasks were perceived as obligations or chores. This framing was done on purpose as we wanted children to deliberately act according to their preferences rather than merely based on social obligations. Nonetheless, we agree that these aspects deserve further discussion in the manuscript:

“It has to be noted that the current study does not contradict previous findings on the collaborative and coordinative skills of young children from various traditional, non-Western populations (6–8,10). Given that children in the current study were unanimously capable of collaborating with their peers when they had to, these findings should not be understood as indicating cultural differences in children’s skills for collaborative problem-solving or collaborative skills per se. In the present study, children across all ages and populations mastered the collaboration game flexibly and skillfully. How children’s social motivation for peer collaboration relates to their coordinative skills to collaborate in more complex and challenging situations than the ones presented here deserves further inquiry (e.g., 10,16). For example, peer collaboration between multiple individuals may insinuate higher demands on social coordination than dyadic interactions (10,15). Further, collaboration which is embedded in subsistence activities and household chores may comprise different affordances than collaboration within playful settings (such as in the current study). Work by Rogoff and colleagues (27,37,26) has emphasized activities for young children in Western, industrialized societies are often framed as playful and non-functional in the context of subsistence activities. If children are asked to contribute within adult activities this may alter the reward value of collaboration notably.” (p. 26)

Further, we have added and modified some information according to the reviewer’s suggestions.

We have added information on children’s access to formal schooling to the method section: 

“Children in this population typically attend institutional childcare and participate in institutionalized education from their second to third year of life onwards. Children were tested in local kindergartens or primary schools. 40 Hai||om dyads (n = 80, 40 girls, MAge = 6.11 years, SDAge = 1.02, MAge Difference = 0.68) from two rural villages in Northern Namibia also participated in the study. Children in these villages attend local primary schools from around six years of age onwards. Finally, 40 Ovambo dyads (n = 80, 38 girls, MAge = 5.73 years, SDAge = 1.26, MAge Difference = 0.37) were tested in a small town in Northern Namibia. All Ovambo children attended either a local kindergarten (typically starting from around two to three years of age) or a local primary school.” (p. 9)

We have further tempered our conclusions regarding cultural “universals” and refer to “regularities” throughout the document (in reference to Rogoff, 2003).

(6)

“Why were the ages chosen? There are major developmental transitions, notably a qualitative shift in Theory of Mind in the age range tested (e.g., False Belief). Could analysis include a look into these issues?”

The age range in the current study has been selected for several reasons. First, children from age three onwards are known to master peer collaboration flexibly and able to choose between individual and social endeavors (e.g. Rekers et at. 2011). Second, the age window was chosen given that children above this age would have likely perceived the collaboration task as too easy and may have thus lost motivation to engage in the tasks throughout the study. Third, this age range reflects the age range in which (urban, Western) children’s Theory of Mind undergoes significant changes which can be captured using the task battery by Wellman & Liu (2004). Finally, given the limited number of children in the Namibian populations, we relied on an opportunistic sampling method including a broad age range to ensure sufficient statistical power to detect potential effects of the predictors.

We concur with Wellman’s and Liu’s (2004) perception that the development of children’s Theory of Mind (including false belief and other facets) needs to be conceived as a gradual, rather than a stepwise process. To ensure that the current results can be interpreted across the age range tested, we included age as a fixed effect into our statistical analyses and discuss such effects more thoroughly in the current version of the manuscript.

“In both Namibian populations, children were recruited via an opportunity sampling approach to maximize the number of participants in the respective communities. German children were recruited from an online database after obtaining written consent from their parents. We chose to assess children within this age range to ensure that children would be capable of collaborating flexibly with their peers while being sufficiently motivated and challenged by the endeavor.” (p. 8)

(7)

“I thought using ToM as a dependent for expressed emotion (itself a proxy for social motivation) was interesting, but also surprising. Could the authors talk a little more about their idea here? Typically, ToM is including as an independent variable. I see the link in the introduction between ToM and social motivation, but in the age ranges here, where at least some of the ToM skills tested are already in place in many three year olds, the relation could go in the other direction.”

We agree that the current study design does not allow us to draw causal conclusions regarding the link between social motivation and Theory of Mind. We conceptualized our analyses as we did since the study was largely inspired and motivated by the framework of Chevallier and colleagues (2012). In their “social motivation theory of autism”, these authors propose a causal link between children’s social motivation as a driving force in the ontogeny of social cognition/Theory of Mind. Accordingly, children’s social-cognitive are developmentally rooted in children’s interactive experience, which is itself facilitated by their motivation to interact and collaborate. Of course, such links should not necessarily be perceived as monocausal, since children’s social motivation may also be strongly shaped by their social-cognitive skills (e.g., social interactions may be perceived as more rewarding if both individuals manage to align their thoughts and beliefs in a mutually beneficial way). 

We account for this aspect in the revised version of the manuscript.

“According to the Social Motivation Theory of Autism (2), but also frameworks posited by other scholars (17,41), children’s tendency to seek and like social interactions (i.e., their social motivation) functions as an ontogenetic foundation for social-cognitive development: Children who are motivated to engage with others more frequently and persistently and thus spend more time in interactions in which they learn to understand and predict others’ behaviors based on social-cognitive inferences. Collaborative endeavors, in which different partners pursue a joint goal, comprise a particularly important context in this regard given that social interdependence and the necessity to coordinate urge individuals to track the beliefs, thoughts, and actions of their collaborative partners (3).” (p. 7)

“We assessed children’s Theory of Mind in order to test whether social motivation would predict children’s Theory of Mind across diverse populations. It has to be noted that the current study design does not allow us to draw definite conclusions with regard to the causality of this link. The study was designed to test predictions put forward in the Social Motivation Theory of Autism (2) across diverse populations. In this framework, social motivation is conceived as an ontogenetic driver of young children’s social-cognitive development. In support of this notion, Burnside and colleagues (42) found young children’s social orientation, another proxy of social motivation, to relate to their performance on an (implicit) false belief task. While the current study thus yields initial support that such a link between social motivation and Theory of Mind may indeed be a cross-cultural regularity, longitudinal research is highly needed to address the causality of this proposed relation.” (p. 24)

(8)

“As noted, perhaps some of the resources in the conclusion can be brought up in the introduction.”

We now point at some of the issues raised in the discussion at an earlier stage in the manuscript:

“Young children across diverse populations are capable and competent in collaborating with their peers for mutual benefit (4–12). In urban, Western societies, children start to collaborate under adult supervision within their second year of life and transfer this skill to successfully master peer interactions in the years to follow (4,13). In such contexts, peer collaboration often takes place in dyadic, playful settings in which children are encouraged to autonomously choose their social partners based on their preferences (14,15). In societies in which social relatedness is prioritized over autonomy, children’s proclivity for collaboration is no less ubiquitous. Some studies suggest that children from rural non-Western, traditional populations may be even more skilled in coordinating and solving collaborative tasks than their urban Western counterparts (6,10,16). For example, Rogoff and colleagues showed that children of Mexican descent growing up in the U.S. outperform their counterparts of European descent in some coordinative aspects of peer collaboration (10,16). That is, children with Mexican indigenous backgrounds coordinate their behaviors with their peers’ by building upon the partner’s actions fluidly and non-verbally. Children of European descent do so more often by relying on verbal coordination and parallel engagement. While these studies thus indicate cultural differences in how children collaborate with their peers, it is without much doubt that the mastery of peer collaboration constitutes a central task in young children’s social development (4,17,18).“ (p. 3)

 (see also point (7) above)

(9)

“Perhaps mention Namibia in the abstract.”

We have added this information to the descriptions of the populations in the abstract.

 

Reviewer 2 wrote:

(1)

“The authors should mention whether children within a dyad were familiar with each other or not.”

We have added this information to the method section:

“Children within each dyad were familiar with each other.” (p. 9)

(2)

“Given that children were allowed to talk prior to their choices (line 252), how would the authors interpret children’s “coordinated decision” to pick individual devices? This is important especially when both the individual and the collaborative devices provide the same material reward. In other words, I urge the authors to explain and speculate whether there is any meaningful difference between a mutual decision to go on separate paths (individual device) and a mutual decision to take one path (collaborative device). Adding more information about the nature and frequencies of children’s communication may help.” 

We thank the reviewer for this suggestion and have added a paragraph to the manuscript in which we discuss this aspect. Indeed, a mutual decision to approach the individual option during forced-choice trials may also be agreed upon jointly. We now discuss this matter in the manuscript. Here, we further want to note that our approach to code only children’s decisions to approach the collaborative option as “collaboration” is rather conservative and in line with other studies on collaborative decision making (e.g., Duguid et al., 2014.).

“We applied a conservative coding scheme to rate children’s preferences during forced-choice trials. That is, children’s choices were only coded as collaboration if they approached the collaboration device. We followed this conservative approach based on previous work on children’s collaborative decision making (e.g., 67). This approach is in line with the current definition of collaboration as a joint endeavor in which two or more individuals engage to pursue a common goal. However, children’s joint decisions to pull individually may also reflect a social commitment. In eight of the 40 German dyads both children pursued the individual option, in contrast to only 17 Hai||om dyads and 14 Ovambo dyads. At the same time, 19 German dyads and 15 Ovambo dyads, but only four Hai||om dyads decided to jointly pursue the collaboration device. Taken together, children agreed on one or the other option in most trials, even though Hai||om showed greater autonomy in their preferences within dyads. Again, this pattern may reflect cultural schemas regarding autonomy and social interdependence which are often reported for foraging societies, such as the Hai||om (e.g., 36).” (p. 23)

 (3)

 “2.1. Model 1 assess children’s motivation for collaboration, but it was a bit unclear what was exactly included in the model: a. Were all the possible interactions between population, condition, reward and age included? The authors report that interactions were tested (line 313), but they should detail exactly what was tested and what was found per model (i.e., non-sig 4-way interaction, non-sig 3-ways, non-sig 2-ways, etc.). […]”

In our analyses we have tested for 2-way interactions between the predictors before reducing model complexity in case likelihood ratio tests did not indicate interactions between predictors. We now describe our approach in more detail in the manuscript and provide model parameters in the supplementary materials.

“During this step, we tested for two-way interactions between predictors but excluded interaction terms if they did not reach statistical significance (detailed results of these analyses are given in the Supplemental Materials).” (p. 15)

(4)

“[…] b. Lines 333: following the previous comment, what was the “preliminary analysis” in which no interactions were found with “reward”?”

(see also point (3) above)

 (5)

“[…] c. Line 362: The authors report an interaction between population and condition, but do not report the results of statistical tests that were done per population (i.e., was the difference among the German children is significant?).” 

In our original analyses we did not run separate analyses for each population. This was done to avoid an inflation of type-I errors due to multiple testing and since the information we are interested in (e.g., whether populations would differ in their effect of condition) is investigated in the models including data from all three populations. While we thus do not give this information in the manuscript, we have added separate analyses for each population to the to the supplemental materials. We have further added full-null-model comparisons to our analyses to avoid inflation of type-I errors (Forstmeier & Schielzeth, 2011).

“Model 1 – Separate Analyses per Subsample

Full Model 1_Subsamples = lmer (Affect ~ Condition + z.Age + Reward + 

z.PositioninVideoLR + z.Sex + z.CollabColor + z.trial + 

(1 + Condition.I + Reward.L || Dyad) + 

(1 + Condition.I + Reward.L || ID) + 

(0 + z.trial | Dyad) + (0 + z.trial | ID) + (1 | trial.id) )

German Subsample: Fixed Effects of Predictors

 LRT df p

Condition 43.96 1 < .001

Age 0.01 1 .069

Reward 10.39 1 .001

*meanHigh Reward = 1.53; meanLow Reward = 1.36

Hai||om Subsample: Fixed Effects of Predictors

 LRT df p

Condition 0.85 1 .358

Age 0.01 1 .927

Reward 2.43 1 .119

*meanHigh Reward = 1.37; meanLow Reward = 1.31

Ovambo Subsample: Fixed Effects of Predictors

 LRT df p

Condition 13.85 1 < .001

Age 0.84 1 .359

Reward 3.48 1 .062

*meanHigh Reward = 1.43; meanLow Reward = 1.35” (supplementary materials)

 (6)

 “[…] d. Line 369: The authors report that overall, children expressed more positive emotions when played with high-value rewards. It is important to know whether a M.E for “reward” was also present in each population separately (“manipulation check”).”

We have addressed whether the effect of reward varies across populations by testing for a 2-way interaction of both predictors in the full sample. As this interaction does not lend support for claims on cultural variation in children’s response to different rewards (Population*Reward: χ2 (2) = 2.73, p = .255), we argue for a culturally recurrent effect of reward: In each population, children expressed more positive emotions during high reward trials as compared to low reward trials (German: meanHigh Reward = 1.53, meanLow Reward = 1.36; Hai||om: meanHigh Reward = 1.37; meanLow Reward = 1.31 ; Ovambo: meanHigh Reward = 1.43; meanLow Reward = 1.35). Applying inferential tests for each subsample separately does, however, not yield statistically significant differences in either population (German: χ2 (1) = 10.39, p = .001; Hai||om: χ2 (1) = 2.43, p = .119; Ovambo: χ2 (1) = 3.48, p = .062). In the manuscript, we briefly refer to these additional analyses presented in in the supplementary materials on p. 17.

(7)

“2.2. Model 2 assess children’s actual preferences for the individual or collaborative devices. a. Can the authors explain why the “affect index” (from model 1) was not included as a predictor in model 2? If “affect index” will be included, then the authors can show a direct link between “positive emotions” during “individualistic or collaborative” tasks and children’s actual choices to engage in individualistic or collaborative tasks. This could shed light on differences at the population level as well as the individual level, for example, even though Ovambo children showed no significant preference for individualistic or collaborative tasks, it is plausible that those with a positive “affect index” favored the collaborative device whereas those with a negative “affect index” favored individual device.”

Following the Reviewer’s suggestion, we have added an additional, explorative analysis to the results in which we included children’s affect index as an additional predictor to model 2. The results of this analysis indicate an interesting pattern: The effect of children’s positive affect during collaboration trials on their subsequent forced-choice preferences depends on age. That is, younger children’s preferences to collaborate or act individually are predicted by the degree to which they expressed more positive emotions in either condition. We report this analysis in the results section and elaborate on it in the discussion:

“To further substantiate the construct validity of both proxies for social motivation in the current study, we also ran an additional exploratory analysis in which we added children’s affect index as a predictor of children’s preferred choices to the model. This analysis revealed that the link between children’s affect index and their subsequent preferences varied with age (Affect Index*Age: χ2 (1) = 4.83, p = .028). To further elaborate on this interaction, we visualized it by plotting the effect of affect index on children’s preferential choices for three distinctive subsamples consisting of children younger than 5.5 years of age (nYoung = 92), those between 5.5 and 6.5 years of age (nIntermediate = 56), and those older than 6.5 years (nOld = 92; see Fig. 5). From these plots, it appears that the link between both proxies for social motivation is apparent among younger children who are more likely to prefer collaboration if they have expressed more positive emotions during previous collaboration trials. With increasing age, however, the two proxies for social motivation become dissociated. Notably, including children’s affect index into the model did not alter the pattern of results described above, as the interaction between population and age remained statistically significant (Population*Age: χ2 (2) = 9.07, p = .011).

Fig 5. Preferential choices. Effect of positive expressed emotions selectively during collaboration trials on preferential choices across populations; affect indexes > 1 indicate that children express more positive emotions during collaboration, as compared to individual trials (fitted values of Generalized Linear Models containing the same Predictors and Controls as the original model are plotted (bold lines) together with 95%-CIs (dotted lines); data points are jittered on the y-axis for visualization purposes)” (p. 18)

“The current study also revealed an interesting pattern regarding the construct validity of both proxies for social motivation investigated here: Younger children’s positive expressed emotions selectively during collaboration trials predicted their subsequent preferences. That is, these children were more likely to pick the collaborative option if they had displayed more positive emotions during previous trials. This pattern was not evident among older children, where both proxies for social motivation were dissected. Since our study was motivated by the Social Motivation Theory of Autism (2), both proxies for social motivation were implemented to fully capture the reward value of social interactions. In accordance with this research (2,71,72), we focused both on children’s liking of collaboration (e.g., their positive expressed emotions) and their seeking of such (e.g., their preferential choices). Interestingly, the scientific literature on rewards emphasizes that seeking and liking reflect dimensions that complement each other, but which are not necessarily linked on an individual level (71). In light of the current study, it is not surprising that children did not necessarily seek what they liked. While younger children may choose their preferred option according to what they like, older children may encounter a more complex decision. At this age, societal norms, such as those structuring peer collaboration and action autonomy, become increasingly relevant for young children (e.g., 73). Depending on populations, some children may have thus sought the individual option to act autonomously and without peer support, whereas others may have preferred to collaborate instead to fulfill normative expectations regarding their willingness to engage with peers in dyadic collaboration.” (p. 26)

(8)

“[…] b. Along the same lines, how can the authors explain Hai||om children’s similar motivation for individual and collaborative devices (model 1, line 365) but their significant preference for actually choosing the individual device? (e.g., Did Hai||om children who are motivated to collaborate (i.e., high “affect index”) inhibited their motivation and chose the individual device (i.e., forced-choice) in order to “achieve autonomy”?).” 

(see also point (8) above)

 (9)

 “[…] c. Figure 4 is interesting. Still, it will be informative to present the actual data points across the age range (so we can get an impression of how many children were per age group), as well as the CIs for the regression lines. To achieve this, I think three separate graphs per population will be more adequate.”

We have modified the figure according to the Reviewer’s suggestions.

Fig 4. Preferential choices. Effect of age on choices (fitted values of Generalized Linear Models containing the same Predictors and Controls as the original model are plotted (bold lines) together with 95%-CIs (dotted lines); data points are jittered on the y-axis for visualization purposes)” (p. 18)

(10)

 “2.3. Model 3 assess children’s ToM scores on the basis of their “affect index” (model 1) and forced-choice (model 2). a. I found the logic behind this model a bit strange, and recommend the authors to better explain it. Specifically, the most valuable thing to predict in my opinion would be children’s forced-choice on the basis of their “affect index” and ToM score. For example, if a given population promotes collaboration (via cultural norms, as the authors suggest), then ToM is not needed at all as collaboration can be also achieved 1) via mere compliance to norms (as was found in line 397), or 2) via explicit verbal coordination (see point 1.3 above), and 3) as also seems to be the case among young Ovambo children (who collaborate even with low ToM scores).”

We have added a paragraph to the introduction in which we explain the logic behind our approach in more detail. Our study design was motivated to test the predictions put forward by Chevallier and colleagues in their Social Motivation Theory of Autism (2012). In their framework, these researchers argue that the early development of social cognition (incl. Theory of Mind) is built upon young children’s social motivation. While we agree that our study design does not allow us to test the causality of the proposed link between the two phenomena, this logic is in line with previous research within this framework (see below).

“According to the Social Motivation Theory of Autism (2), but also frameworks posited by other scholars (17,41), children’s tendency to seek and like social interactions (i.e., their social motivation) functions as an ontogenetic foundation for social-cognitive development: Children who are motivated to engage with others more frequently and persistently and thus spend more time in interactions in which they learn to understand and predict others’ behaviors based on social-cognitive inferences. Collaborative endeavors, in which different partners pursue a joint goal, comprise a particularly important context in this regard given that social interdependence and the necessity to coordinate urge individuals to track the beliefs, thoughts, and actions of their collaborative partners (3). Studies have provided initial evidence that social motivation may indeed be linked to Theory of Mind development among urban Western children (e.g., 42). Yet, it remains unclear whether this link can be applied outside such populations. First, recent research has revealed that children’s Theory of Mind skills vary considerably across cultures (e.g., 43–47). Moreover, other studies showed that psychological correlates of Theory of Mind typically observed among Western populations do not necessarily persevere outside such samples. For example, while the number of siblings is a well-documented predictor of Theory of Mind among Western children (48), this effect is not evident among Iranian children (46,49,50). Similarly, authoritarian parenting practices are negatively linked to U.S.-American children’s Theory of Mind acquisition, but such links are absent among Korean children tested in the same study (50, see also 51).” (p. 7)

We further note that children’s preferences to collaborate did not necessarily rely upon sophisticated Theory of Mind skills. First, all children in the current study were clearly capable of collaborating with their peer, as indicated by their behavior throughout the collaboration game more generally. As such, we believe that their tendency to prefer collaboration or individual activity was indicative of a motivational disposition, rather than a socio-cognitive skill. To avoid confusion about the proposed links, we discuss this aspect now more thoroughly in the manuscript

“We assessed children’s Theory of Mind in order to test whether social motivation would predict children’s Theory of Mind across diverse populations. It has to be noted that the current study design does not allow us to draw definite conclusions with regard to the causality of this link. The study was designed to test predictions put forward in the Social Motivation Theory of Autism (2) across diverse populations. In this framework, social motivation is conceived as an ontogenetic driver of young children’s social-cognitive development. In support of this notion, Burnside and colleagues (42) found young children’s social orientation, another proxy of social motivation, to relate to their performance on an (implicit) false belief task. While the current study thus yields initial support that such a link between social motivation and Theory of Mind may indeed be a cross-cultural regularity, longitudinal research is highly needed to address the causality of this proposed relation.” (p. 24)

(11)

“[…] b. In this case, a M.E for population can strengthen the author’s point regarding cultural differences in collaboration that are independent of individual differences (e.g., ToM).”

To show that the including Theory of Mind as a predictor in model 2 does not change the conclusions drawn in the manuscript, we have run the suggested model:

Exploratory Model 2 = glmer (Preferential Choice ~ (z.Age + Affect Index)^2 + Population + ToM + 

 z.Sex +

 (1 | Dyad) )

Results suggest that the effects of our predictors remain unaffected by adding Theory of Mind as a control variable: The interaction between children’s affect index and their age remains statistically significant (χ2 (1) = 4.72, p = .030). Further, the main effect of population remains statistically significant (χ2 (1) = 13.38, p = .001). 

As described above, our model selection was motivated by the work of Chevallier and colleagues (2012) who posit a causal link between social motivation and social cognition. Further, all participating children, regardless of their Theory of Mind skills, successfully collaborated throughout the study. To avoid confusion about the framework underlying the current study, we have decided not to add this suggested analysis into the current manuscript but are happy to include it if requested.

(12)

“[…] c. The authors should also expand on possible reasons for big differences in ToM scores across populations (line 401), especially when such differences were not found in other cultures (e.g., res 41 and 44). It could definitely be that the ToM task that was used here is not suited for Ovambo and Hai||om, but I believe this should be discussed nonetheless.” 

We now offer a more detailed discussion of this argument in the manuscript. We argue that the current study protocol yields a fair assessment of children’s Theory of Mind within populations, but argue that interpreting the cultural differences in children’s Theory of Mind performance in the current study as strong proof for systematic variation in the skill per se is beyond the scope of our investigation.

“It is important to note that we did not have specific predictions regarding cross-cultural differences or homogeneity in Theory of Mind. The current results, however, add to a growing body of research documenting substantial variation in children’s (explicit) Theory of Mind across populations (43–45). It is most likely that a combination of different factors may underlie the cross-cultural variability observed here: To achieve high scores on the current set of verbal Theory of Mind tasks, German children may benefit from the more frequent conversations about mental states they experience in their daily lives (38). One practice that may be particularly influential in this regard is parental mind-mindedness, or their “proclivity to view their children as mental agents” (65, p. 1297; see also 66). In one study, Hughes and colleagues (68) assessed parental mind-mindedness in adults from Hong Kong and the U.K. in addition to their children’s Theory of Mind skills. While these researchers documented cultural variation in either variable, both phenomena were linked within populations. It thus appears that parenting practices may underlie developmental variation in Theory of Mind across populations, while also relating to interindividual variation within populations.” (p. 25)

---

## [Decision Letter · Decision Letter 1]

27 Oct 2020

Cultural Variation in Young Children’s Social Motivation for Peer Collaboration and its Relation to the Ontogeny of Theory of Mind

PONE-D-20-07321R1

Dear Dr. Stengelin,

We’re pleased to inform you that your manuscript has been judged scientifically suitable for publication and will be formally accepted for publication once it meets all outstanding technical requirements.

Kind regards,

Valerio Capraro

Academic Editor

PLOS ONE

Additional Editor Comments (optional):

Reviewers' comments:

Reviewer's Responses to Questions

**Comments to the Author**

1. If the authors have adequately addressed your comments raised in a previous round of review and you feel that this manuscript is now acceptable for publication, you may indicate that here to bypass the “Comments to the Author” section, enter your conflict of interest statement in the “Confidential to Editor” section, and submit your "Accept" recommendation.

Reviewer #1: All comments have been addressed

2. Is the manuscript technically sound, and do the data support the conclusions?

Reviewer #1: Yes

3. Has the statistical analysis been performed appropriately and rigorously? 

Reviewer #1: Yes

4. Have the authors made all data underlying the findings in their manuscript fully available?

Reviewer #1: Yes

5. Is the manuscript presented in an intelligible fashion and written in standard English?

Reviewer #1: Yes

6. Review Comments to the Author

Reviewer #1: I thought this was a very responsive revision and the paper is now much clearer and more informative. I applaud the authors on their work.

7. PLOS authors have the option to publish the peer review history of their article (what does this mean?). If published, this will include your full peer review and any attached files.

Reviewer #1: No

---

## [Editor Report · Acceptance letter]

4 Nov 2020

PONE-D-20-07321R1 

Cultural Variation in Young Children’s Social Motivation for Peer Collaboration and its Relation to the Ontogeny of Theory of Mind 

Dear Dr. Stengelin:

I'm pleased to inform you that your manuscript has been deemed suitable for publication in PLOS ONE. Congratulations! Your manuscript is now with our production department. 

Kind regards, 

on behalf of

Dr. Valerio Capraro 

Academic Editor

PLOS ONE